# A Conflict-aware Evidential Framework for Reliable Sleep Stage Classification

**Yunzhi Tian** [1]  **Dekui Wang** [1]  **Qirong Bu** [1]  **Wei Zhou** [1]  **Xingxing Hao** [1]  **Jun Feng** [1]

## Abstract

Multi-view learning has been widely applied for sleep stage classification using multi-modal data. However, existing methods typically assume that different modalities are well-aligned, which is often unattainable in real-world scenarios, thereby compromising the reliability of the staging results. In this paper, we propose ConfSleepNet, a conflict-aware evidential framework that dynamically resolves inter-view conflicts. The framework consists of multi-view evidence extraction and conflict-aware aggregation. In the first phase, it learns category-related evidence from different modalities, which represents the degree of support for individual sleep stages. Considering the inherent characteristics of varying modalities, we propose hybrid category structures for different modalities to promote more reasonable evidence learning. In the second phase, view-specific opinions, including prediction results and uncertainty, are constructed from the learned evidence. Notably, we propose a novel conflict-aware aggregation method that integrates these view-specific opinions into a reliable joint decision. This mechanism can effectively resolve conflicts among opinions and synthesize them into a reliable joint decision. Both theoretical analysis and experimental results demonstrate the effectiveness of ConfSleepNet in sleep staging tasks. The code is available at ConfSleepNet.

## 1. Introduction

Sleep occupies approximately one-third of a human's lifetime and is crucial for maintaining mental and physical well-being (Lane et al., 2023). Unfortunately, an increasing number of people suffer from sleep disorders, posing significant public health challenges (Perez-Pozuelo et al., 2020; Grandner, 2022). For example, about 36% of the global population and 176 million Chinese experience sleep disorders, leading to health issues such as cardiovascular diseases, cognitive decline, and memory deterioration. Clinically, sleep staging is a fundamental process for human sleep assessment (Kong et al., 2023). Traditionally, this task is performed manually by sleep experts based on overnight polysomnography (PSG), a process that typically takes several hours (Portier et al., 2000). In contrast, machine-assisted classification models can accomplish this task within seconds. Therefore, automating the sleep scoring process is imperative.

Previous work on automatic sleep staging can be divided into single-view methods (Supratak et al., 2017; Li et al., 2024; Phyo et al., 2023) and multi-view methods (Phan et al., 2022; Chen et al., 2023; Pradeepkumar et al., 2024) based on the type of network input. Single-view methods typically use the single-modal information provided by electroencephalogram (EEG) for sleep staging. In contrast, some studies have explored the introduction of additional modalities, such as electrooculogram (EOG), to provide complementary view information for the classification model. Notably, existing methods often assume that different views are well-aligned, assigning equal weights to different views (e.g., via concatenation) or learning a fixed weight for each view (Phan et al., 2022; Pradeepkumar et al., 2024). However, this assumption is not always valid in real-world applications, as information from different views may conflict (e.g., two views may point to different sleep stages). Therefore, a well-designed model must be aware of such conflicts and dynamically adjust the importance of each view during decision-making.

In this work, we propose ConfSleepNet, a conflict-aware evidential framework that effectively manages inter-view conflicts and promotes reliable sleep staging decisions. Specifically, ConfSleepNet consists of two main phases: multi-view evidence extraction and conflict-aware aggregation. First, view-specific evidential deep neural networks (DNNs) are employed to learn category-related evidence from multi-modal inputs, including EEG, EOG, and their combination. Notably, unlike existing methods that treat all modalities indiscriminately using a five-class classification design, we adopt a hybrid category structure combining coarse-grained and fine-grained categories. This enables the model

[1]College of Computer Science, Northwest University, Xi'an, China. Correspondence to: Dekui Wang <dekui_wang@nwu.edu.cn>, Jun Feng <fengjun@nwu.edu.cn>.

*Proceedings of the 43rd International Conference on Machine Learning*, Seoul, South Korea. PMLR 306, 2026. Copyright 2026 by the author(s).

to extract evidence from each view in a reasonable and physiologically aligned manner. In the multi-view aggregation phase, we construct view-specific opinions that incorporate class belief masses and prediction uncertainties using Dirichlet distributions parameterized by the learned evidence. For final decision-making, we propose a novel conflict-aware multi-view aggregation method that explicitly accounts for inter-view conflicts, thereby integrating multiple view-specific opinions into a reliable joint opinion.

Our main contributions are summarized as follows: (1) We propose an evidential framework named ConfSleepNet, which introduces a hybrid category structure to enable differentiated evidence learning from EEG and EOG signals. (2) We present a conflict-aware multi-view aggregation method that enhances the reliability of classification results by explicitly considering inter-view conflicts. The theoretical analysis in the Section 3.5 demonstrates that this method can integrate multiple potentially conflicting opinions into a reasonable joint opinion. (3) The proposed method is evaluated on multiple public datasets, and the results show that it outperforms state-of-the-art baseline methods.

## Conflict of Interest Disclosure

The authors declare that they have no financial conflicts of interest related to this work.

## 2. Related Work

### 2.1. Automatic Multi-View Sleep Staging

Leveraging multi-view data for learning can provide richer information compared to single-view data, and its effectiveness in sleep staging tasks has been well demonstrated. Existing methods typically learn features independently from each view, followed by feature-level fusion. For example, Chambon et al. (2018) learn high-level representations from PSG signals and subsequently construct a joint representation for classification. Similarly, Phan et al. (2018) convert raw signals into time-frequency images and then perform feature-level fusion for prediction. These works adopt simple operations such as concatenation for multi-view fusion, implicitly assuming that all views are equally important. However, this assumption does not always hold, raising concerns about model reliability. For this issue, Jia et al. (2021) employ a dual-stream network to extract features independently from EEG and EOG signals, and leverage an attention mechanism for feature fusion. Furthermore, Dai et al. (2023) adopt a Transformer encoder (Vaswani, 2017) for view-specific feature extraction and multi-view feature fusion based on a self-attention mechanism. Although such methods can appropriately fuse multiple views, they remain unable to detect potential noise within each view, thereby compromising the robustness of the final predictions.

### 2.2. Conflicting Multi-View Learning

Early multi-view learning works relied on Bayesian methods (Neal, 2012; Gal & Ghahramani, 2016) to construct weight distributions but were limited by high computational costs. Subsequently, ensemble methods (Egele et al., 2022; Ganaie et al., 2022) advanced the field by combining predictions from multiple independent sub-networks. Despite making progress, these approaches overlooked inter-view conflicts. In recent years, evidential deep learning (EDL) (Sensoy et al., 2018) has achieved significant success in multi-view learning tasks (Li et al., 2023; Xia et al., 2024). Within the EDL framework, related works (Han et al., 2022; Shao et al., 2024; Huang et al., 2025) have employed Dempster-Shafer theory (Dempster, 1968) to assign lower weights to views with high uncertainty, thereby addressing conflicts among views. However, existing EDL-based methods suffer from a limitation in handling the influence of conflicts on prediction uncertainty: they implicitly assume that incorporating more certain opinions will always reduce overall uncertainty (Liu et al., 2022; Zhang et al., 2023; Xu et al., 2024). We consider this assumption unreasonable and consequently propose a conflict-aware multi-view aggregation method, along with a theoretical analysis of its advantages.

## 3. The Proposed Method

The proposed ConfSleepNet handles a sequence of 30-s sleep epochs $\mathbf{x}^{(L)} = \{x_1, x_2, ..., x_L\}$ and classifies each epoch $x_i$ into sleep stage $y_i \in \{W, N1, N2, N3, REM\}$ following the AASM rule (Berry et al., 2012). Herein, $x_i \in \mathbb{R}^{2 \times T}$ contains EEG and EOG signals with equal sampling frequency, and $T$ is the number of points in an epoch. The overall architecture of ConfSleepNet is illustrated in Fig. 1, with further detailed discussions to be presented in the following subsections.

### 3.1. Design Principles

EEG and EOG are the two most widely used physiological signals in clinical sleep staging applications. For these two different modalities, their signal characteristics are both distinctive and physiologically complementary. Therefore, their predictions for the same sample may be either consistent or conflicting. We argue that a sleep staging model should possess good robustness, enabling reliable final decision-making even when predictions from different views are in conflict. To achieve this goal, the following design principles aim to incorporate both complementarity (i.e., EEG and EOG contain complementary information) and reliability (i.e., reliable multi-view aggregation) into the model.

***Principle 1 (Complementarity):*** *The complementary nature of EEG and EOG is beneficial for sleep stage classifica-*

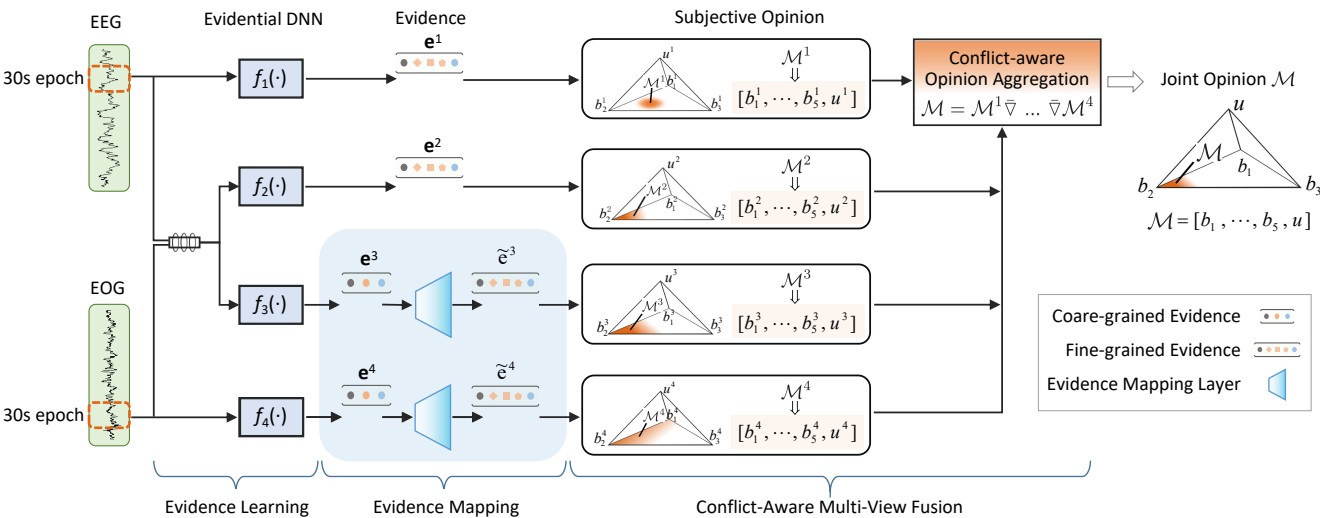

*Figure 1.* Illustration of ConfSleepNet. Four evidential DNNs $\{f_v(\cdot)\}_{v=1}^{4}$ learn class-specific evidences from various views, which involve two different category structures. Next, an evidence mapping layer maps the coarse-grained evidence into fine-grained evidence. After that, we construct view-specific opinions $\{\mathcal{M}^v\}_{v=1}^{4}$ based on the obtained evidence and then combine them to form a joint opinion $\mathcal{M}$. In multi-view fusion, we identify conflictive opinions via uncertainty estimation and accordingly reduce their impact in decision-making.

*tion.* According to the AASM standard, PSG signals are typically segmented into 30-second epochs and classified into one of five sleep stages: wake ($W$), rapid eye movement ($REM$), and three non-REM ($NREM$) stages ($N1$, $N2$, and $N3$). Previous works have often treated the two modalities (i.e., EEG and EOG) indiscriminately by directly performing five-class classification on them. However, in clinical practice, EEG serves as the primary modality for sleep staging, providing discriminative neurophysiological patterns for different sleep states. Meanwhile, EOG, as an important auxiliary modality to EEG-based sleep staging, plays a key role primarily in identifying the $REM$ stage. We provide a detailed description of the signal characteristics of EEG and EOG in Appendix A.1. Based on the signal characteristics of the different modalities, ConfSleepNet designs a coarse-grained classification structure for EOG (i.e., distinguishing $W$, $REM$, and $NREM$) and, in parallel, a fine-grained classification structure for EEG (i.e., further subdividing $NREM$ into three sub-stages: $N1$, $N2$, and $N3$). Through this design, ConfSleepNet is able to fully leverage the complementary characteristics of EEG and EOG, thereby improving model performance.

***Principle 2 (Consistent multi-view aggregation): Incorporating additional consistent view opinions reduces overall uncertainty (i.e., resulting in a more confident opinion).*** How to reasonably integrate multiple views remains an open problem. Recently, Xu et al. (2024) proposed ECML, which represents the current state-of-the-art. The core idea of this method is that integrating an uncertain view into the original view increases the overall uncertainty. For example, consider two views on the same instance, producing opinions $\mathcal{M}^1$ and $\mathcal{M}^2$ respectively. Both opinions yield the same

prediction but have different levels of uncertainty regarding their own predictions ($u^1 = 0.3$ and $u^2 = 0.8$). When aggregating $\mathcal{M}^1$ and $\mathcal{M}^2$ using ECML, the resulting joint opinion has an uncertainty of 0.436, which is higher than $u^1$. We consider this unreasonable because the aggregation result of ECML does not account for the consistency between opinions. Intuitively, incorporating additional consistent views should reduce the overall uncertainty, thereby leading to a more confident final decision.

***Principle 3 (Conflictive multi-view aggregation): Incorporating a conflictive view opinion with higher uncertainty increases the overall uncertainty (i.e., resulting in a less confident opinion).*** In sleep staging, the EEG and EOG views may produce inconsistent opinions for the same sample, which typically arises from factors such as differences in signal characteristics, noise interference, or physiological variability. Therefore, the proposed ConfSleepNet should be robust against conflicting views, which is crucial for clinical adoption. However, most existing methods lack appropriate mechanisms to handle conflicting views, thereby compromising model performance. We argue that incorporating a conflicting opinion with high uncertainty weakens the confidence in the original opinion. Consequently, the proposed multi-view aggregation method should explicitly assess the degree of inter-view conflict while reasonably incorporating the impact of conflict on the final decision.

### 3.2. View-Specific Evidence Learning

ConfSleepNet first transforms the raw input $x$ into high-level feature representations. Specifically, we utilize two parallel convolutional branches with different kernel sizes to

extract time-frequency features. Subsequently, an improved GCNet-1D attention module (Cao et al., 2019) is employed to enhance the features, denoted as $h^l$ and $h^s$. Furthermore, ConfSleepNet applies a cross-attention mechanism to learn the interactive information between $h^l$ and $h^s$, yielding the interacted features $h^{sl}$ and $h^{ls}$. Notably, we introduce a bidirectional long short-term memory layer in this process to model the stage transition patterns across multiple consecutive sleep epochs. Finally, we combine the enhanced features with the interactive features to obtain the high-level feature representation $F$. See the Appendix A.2 for the detailed feature extraction process.

Most existing deep learning methods typically perform classification based on the point-estimated distributions provided by *Softmax*. However, such approaches can still produce overconfident outputs even when confronted with low-quality data, thus exhibiting limitations in conflictive multi-view learning. Evidential deep learning (EDL) addresses this challenge by introducing the evidence framework of subjective logic (Jsang, 2018). Based on EDL, we construct $V$ evidential DNNs $\{f_v(\cdot)\}_{v=1}^V$ to extract evidence from the high-level features $F$ as follows:

$$\mathbf{e}^v = f_v(\cdot) \qquad (1)$$

Unlike most deep learning methods, we use the *SoftPlus* activation function to extract an evidence vector $\mathbf{e}^v$ containing $k$ elements from the features (where each element represents the model's degree of support for each category). Notably, $V = 4$ corresponds to the four designed evidential DNNs, which align with the hybrid category structure in ConfSleepNet. Specifically, the network inputs consist of three forms: single-view EEG, single-view EOG, and multi-view (i.e., a combination of EEG and EOG). $f_1(\cdot)$ and $f_2(\cdot)$ provide fine-grained five-category evidence for the single-view EEG and multi-view inputs, respectively. Meanwhile, $f_3(\cdot)$ and $f_4(\cdot)$ provide coarse-grained three-category (i.e., $W$, $REM$, and $NREM$) evidence for the single-view EOG and multi-view inputs, respectively. Based on the input type, we divide the evidential DNNs into single-view DNNs (corresponding to $f_1(\cdot)$ and $f_4(\cdot)$) and multi-view DNNs (corresponding to $f_2(\cdot)$ and $f_3(\cdot)$). The main components of each network are consistent, and we provide more details on the network architecture in the Appendix A.3.

### 3.3. Evidence Mapping Layer

The evidences $\{\mathbf{e}^v\}_{v=1}^4$ learned by the view-specific DNNs $\{f_v(\cdot)\}_{v=1}^4$ have category structures with different granularities (i.e., $\mathbf{e}^1$ and $\mathbf{e}^2$ are five-class evidence vectors, while $\mathbf{e}^3$ and $\mathbf{e}^4$ are three-class evidence vectors). Before multi-view fusion, we design an evidence mapping layer to convert the coarse-grained evidence vectors $\mathbf{e}^3$ and $\mathbf{e}^4$ into fine-grained evidence vectors $\tilde{\mathbf{e}}^3$ and $\tilde{\mathbf{e}}^4$, respectively. Specifically,, the evidence mapping function is achieved by $\tilde{\mathbf{e}}^v = \mathbf{e}^v \times \mathbf{U}$,

where $\mathbf{U} \in \mathbb{R}^{3 \times 5}$ is a matrix and expected to be

$$\mathbf{U} = \begin{bmatrix} 1 & 0 & 0 & 0 & 0 \\ 0 & \frac{1}{3} & \frac{1}{3} & \frac{1}{3} & 0 \\ 0 & 0 & 0 & 0 & 1 \end{bmatrix}$$

The idea of evidence mapping can be summarized as three points: (1) The total amount of evidence should remain unchanged after the evidence mapping. Hence, each row of matrix $\mathbf{U}$ should sum up to 1. (2) Equivalent mapping of evidence for each class should be guaranteed. This mainly enables two things: First, equivalent evidence mapping for class $W$ and $REM$; second, the total evidences for $N1$, $N2$ and $N3$ in $\tilde{\mathbf{e}}^v$ ($v = 3, 4$) should equal that of $NREM$ in $\mathbf{e}^v$. (3) The evidence for $NREM$ stage should be equally assigned to its sub-stages, $N1$, $N2$ and $N3$. This is because, while learning the evidence for $NREM$, instances of class $N1$, $N2$, and $N3$ have the same distribution. We compare the performance of this mapping strategy with other strategies in the Appendix A.4 and discuss its feasibility.

### 3.4. Conflict-Aware Multi-View Fusion

**Constructing View-Specific Opinions.** After evidence extraction, we then model the distributions of class probabilities by using the Dirichlet distribution. Specifically, the Dirichlet distribution of the $v^{th}$ view is parameterized by $\boldsymbol{\alpha}^v = (\alpha_1^v, ..., \alpha_5^v)$, where $\boldsymbol{\alpha}^v = \mathbf{e}^v + 1$ for $v = 1, 2$ and $\boldsymbol{\alpha}^v = \tilde{\mathbf{e}}^v + 1$ for $v = 3, 4$. This ensures that the Dirichlet distribution is nonsparse. Specifically, for the $v^{th}$ view, the probability density function of the Dirichlet distribution is defined as:

$$D(\boldsymbol{p}^v | \boldsymbol{\alpha}^v) = \begin{cases} \frac{1}{B(\boldsymbol{\alpha}^v)} \prod_{k=1}^5 (p_k^v)^{\alpha_k^v - 1} & for\ \boldsymbol{p}^v \in T_5 \\ 0 & otherwise \end{cases} \qquad (2)$$

where $B(\cdot)$ denotes the multinomial beta function, and $T_5$ is the 5-dimensional unit simplex with the standard and unique definition:

$$T_5 = \left\{ \boldsymbol{p}^v | \sum_{k=1}^5 p_k^v = 1\ and\ 0 \leq p_1^v, ..., p_5^v \leq 1 \right\} \qquad (3)$$

Unlike existing *Softmax*-based methods, which only capture first-order uncertainty, the Dirichlet distribution models higher-order category probabilities, enabling more precise uncertainty estimation.

From the Dirichlet distributions, we can further construct view-specific opinions. The opinion stemmed from the $v^{th}$ view can be described as an ordered tuple $\mathcal{M}^v = (\boldsymbol{b}^v, u^v)$, where $\boldsymbol{b}^v = (b_1^v, ..., b_5^v)$ assigns belief mass to potential sleep stages, and uncertainty mass $u^v$ captures the information of ambiguity or vacuity according to the acquired

evidences. For the $v^{th}$ view, we normalize the evidence to obtain belief and uncertainty masses:

$$b_k^v = \frac{\alpha_k^v - 1}{S^v} \ , \ u^v = \frac{K}{S^v} \tag{4}$$

where $b_k^v$ is the belief mass of the $k^{th}$ category, and $u^v$ is the uncertainty of the opinion, and $K = 5$ is the number of sleep stages. $S^v = \sum_{k=1}^5 \alpha_k^v$ is the strength of Dirichlet distribution. According to subjective logic theory, both $b_k^v \in \boldsymbol{b}^v$ and $u^v$ must be non-negative, and their sum should be equal to 1, that is

$$\sum_{k=1}^5 b_k^v + u^v = 1 \ ( \ b_k^v, \ u^v \in [0, 1] \ ) \tag{5}$$

**Aggregating View-Specific Opinions.** This section presents a conflict-aware multi-view aggregation method (CMAM) for synthesizing a reliable joint opinion from different views $\{\mathcal{M}^v\}_{v=1}^4$. Due to the influence of noise or other factors, opinions formed on different views may diverge. Therefore, following Principles 2 and 3 proposed in Section 3.1, we design CMAM to reasonably aggregate consistent or conflicting opinions. First, we introduce a metric in Definition 1 for quantifying the degree of conflict between two opinions.

**Definition 1 (Conflict-degree metric).** For two opinions $\mathcal{M}^a = (\boldsymbol{b}^a, u^a)$ and $\mathcal{M}^b = (\boldsymbol{b}^b, u^b)$ over the same instance, the degree of conflict between $\mathcal{M}^a$ and $\mathcal{M}^b$ is calculated as:

$$C(\mathcal{M}^a, \mathcal{M}^b) = 1 - \frac{\sum_k b_k^a \cdot b_k^b}{\sum_i b_i^a \cdot \sum_j b_j^b} \tag{6}$$

This metric guarantees two things: (1) $C = 0$ indicates perfectly consistent opinions. This situation arises when two absolute opinions supporting the same category are presented. (2) $C = 1$ indicates completely conflicting opinions. Typically, $C$ ranges from 0 to 1, and a larger $C$ signifies greater conflict. Based on this metric, the details of the proposed CMAM are presented as follows.

**Definition 2 (Conflict-aware multi-view aggregation).** Let $\mathcal{M}^a = (\boldsymbol{b}^a, u^a)$ and $\mathcal{M}^b = (\boldsymbol{b}^b, u^b)$ be two opinions over the same instance. The combined opinion $\mathcal{M}^{a\bar{\nabla}b}$ is calculated as follows:

$$\mathcal{M}^{a\bar{\nabla}b} = \mathcal{M}^a \bar{\nabla} \mathcal{M}^b = (\boldsymbol{b}^{a\bar{\nabla}b}, u^{a\bar{\nabla}b}) \tag{7}$$

$$u^{a\bar{\nabla}b} = C\frac{2u^a u^b}{u^a + u^b} + (1 - C)u^a u^b \tag{8}$$

$$b_k^{a\bar{\nabla}b} = \frac{u^a b_k^b + u^b b_k^a + (1 - C)u^a u^b(b_k^a + b_k^b)}{u^a + u^b} \tag{9}$$

The obtained opinion $\mathcal{M}^{a\bar{\nabla}b}$ is a combination of $\mathcal{M}^a$ and $\mathcal{M}^b$. Its quality not only depends on the quality of the original opinions $\mathcal{M}^a$ and $\mathcal{M}^b$, but also is affected by the degree

of conflict between them. In particular, one opinion gains additional confidence when incorporating a consistent opinion, and becomes uncertain when combining a conflicting opinion. This characteristic distinguishes our proposed CMAM from other multi-view fusion methods, such as ECML (Xu et al., 2024) and DS-based fusion (Han et al., 2022). Following the subjective logic theory (Jsang, 2018), the uncertainty and belief masses of the combined opinion $\mathcal{M}^{a\bar{\nabla}b}$ must be non-negative and sum up to one, i.e., $\sum_k b_k^{a\bar{\nabla}b} + u^{a\bar{\nabla}b} = 1$, and the proof in Appendix A.5.

Following Definition 2, we can further combine more than two opinions with the following rule:

$$\mathcal{M} = \mathcal{M}^1 \bar{\nabla} \mathcal{M}^2 \bar{\nabla} \ ... \ \bar{\nabla} \mathcal{M}^n \ \ (n \geq 2) \tag{10}$$

This is actually an extension of combining two opinions discussed in Definition 2. Based on the above multi-opinion fusion rule, we can get the final multi-view joint opinion, and hence get the probability of each category and the overall uncertainty. Although CMAM improves the reliability of the final prediction, it still has certain limitations, and further details are provided in the Appendix A.6.

### 3.5. Theoretical Analysis

In this subsection, we theoretically analyze the advantages of CMAM. Section 3.1 presents two principles, $Principle\ 2$ and 3, to guide reasonable multi-view aggregation. The following two propositions provide theoretical analysis to support these principles.

**Proposition 1.** *Given an opinion $\mathcal{M}^o = (\boldsymbol{b}^o, u^o)$, integrating additional consistent opinion $\mathcal{M}^a = (\boldsymbol{b}^a, u^a)$ into $\mathcal{M}^o$ would produce a new opinion with lower uncertainty than $u^o$.*

*Proof.*

Let $\mathcal{M}^{o\bar{\nabla}a} = (\boldsymbol{b}^{o\bar{\nabla}a}, u^{o\bar{\nabla}a})$ be the combination of $\mathcal{M}^o$ and $\mathcal{M}^a$. The degree of conflict between $\mathcal{M}^o$ and $\mathcal{M}^a$, denoted by $C$, is approaching 0 since they are consistent. Such that

$$\lim_{C \to 0} u^{o\bar{\nabla}a} = \lim_{C \to 0}(1 - C)u^o u^a + C\frac{2u^o u^a}{u^o + u^a}$$
$$= u^o u^a < u^o$$

Proposition 1 demonstrates that CMAM can enhance prediction confidence when incorporating consistent opinions.

**Proposition 2.** *Given an opinion $\mathcal{M}^o = (\boldsymbol{b}^o, u^o)$, integrating a conflictive opinion $\mathcal{M}^b$ with higher uncertainty (i.e., $u^b > u^o$) into $\mathcal{M}^o$ would increase the uncertainty.*

*Proof.*

Let $\mathcal{M}^{o\bar{\nabla}b} = (\boldsymbol{b}^{o\bar{\nabla}b}, u^{o\bar{\nabla}b})$ be the combination of $\mathcal{M}^o$ and $\mathcal{M}^b$. Since $\mathcal{M}^o$ and $\mathcal{M}^b$ are in conflict, the degree of con-

*Table 1.* Performance comparison between ConfSleepNet and state-of-the-art methods for automatic sleep staging. ↑ means higher is better. The best results are highlighted in bold.

| DATASET | METHOD | OVERALL METRICS | | F1-SCORE FOR EACH CLASS | | | | |
|---|---|---|---|---|---|---|---|---|
| | | ACC ↑ | MF1 ↑ | W | N1 | N2 | N3 | REM |
| SLEEPEDF-20 | DEEPSLEEPNET (SUPRATAK ET AL., 2017) | 82.0 | 76.9 | 84.7 | 46.6 | 85.9 | 84.8 | 82.4 |
| | TINYSLEEPNET (SUPRATAK & GUO, 2020) | 85.4 | 80.5 | 90.1 | 51.4 | 88.5 | 88.3 | 84.3 |
| | SALIENTSLEEPNET (JIA ET AL., 2021) | 86.3 | 80.6 | 90.8 | 49.9 | 89.0 | 84.8 | 88.4 |
| | XSLEEPNET (PHAN ET AL., 2022) | 86.4 | 80.9 | 91.5 | 51.1 | 87.5 | 86.9 | 87.4 |
| | SLEEPYCO (LEE ET AL., 2024) | 86.3 | 80.6 | 89.1 | 50.3 | 88.3 | 87.0 | 88.5 |
| | FLEXIBLESLEEPNET (REN ET AL., 2025) | 86.8 | 81.4 | 91.8 | **53.2** | **89.8** | 85.0 | 87.2 |
| | TMCEK (LIANG ET AL., 2025) | 85.0 | 80.2 | 87.0 | 50.3 | 87.5 | 88.1 | 88.1 |
| | HMDT-NET (WANG ET AL., 2026) | 86.4 | 80.5 | 89.1 | 48.5 | 89.2 | 87.3 | 88.4 |
| | CONFSLEEPNET- | 86.3 | 81.3 | 91.2 | 49.4 | 88.0 | 90.2 | 87.8 |
| | CONFSLEEPNET | **87.2** | **81.8** | **91.9** | 49.5 | 88.2 | **90.4** | **89.1** |
| SLEEPEDF-78 | DEEPSLEEPNET (SUPRATAK ET AL., 2017) | 77.8 | 71.8 | 90.9 | 45.0 | 79.2 | 72.7 | 71.1 |
| | TINYSLEEPNET (SUPRATAK & GUO, 2020) | 83.1 | 78.1 | 92.8 | **51.0** | 85.3 | 81.1 | 80.3 |
| | SALIENTSLEEPNET (JIA ET AL., 2021) | 82.6 | 76.5 | 92.3 | 50.5 | 84.4 | 71.2 | 84.2 |
| | XSLEEPNET (PHAN ET AL., 2022) | 84.0 | 78.7 | 92.6 | 50.3 | 85.5 | 79.2 | 85.7 |
| | SLEEPYCO (LEE ET AL., 2024) | 84.6 | 78.7 | 92.4 | 50.4 | 86.0 | 80.5 | 84.2 |
| | FLEXIBLESLEEPNET (REN ET AL., 2025) | 84.6 | 78.1 | 92.1 | 48.2 | 85.7 | 80.7 | 83.9 |
| | TMCEK (LIANG ET AL., 2025) | 81.4 | 77.5 | 92.2 | 47.5 | 84.2 | 79.6 | 84.3 |
| | HMDT-NET (WANG ET AL., 2026) | 84.5 | 77.9 | 92.5 | 50.3 | 84.4 | 81.2 | 80.9 |
| | CONFSLEEPNET- | 84.2 | 78.2 | 92.6 | 45.4 | 85.6 | 82.4 | 85.1 |
| | CONFSLEEPNET | **85.3** | **78.8** | **93.1** | 45.8 | **86.1** | **82.8** | **86.0** |
| MASS-SS3 | DEEPSLEEPNET (SUPRATAK ET AL., 2017) | 86.2 | 81.6 | 87.3 | 59.8 | 90.3 | 81.5 | 89.3 |
| | TINYSLEEPNET (SUPRATAK & GUO, 2020) | 87.5 | 83.2 | 87.3 | **62.7** | 91.8 | 85.5 | 88.6 |
| | XSLEEPNET (PHAN ET AL., 2022) | 86.9 | 82.7 | 89.5 | 60.3 | 90.1 | 83.6 | 89.8 |
| | SLEEPYCO (LEE ET AL., 2024) | 86.8 | 82.5 | 89.2 | 60.1 | 90.4 | 83.8 | 89.1 |
| | TMCEK (LIANG ET AL., 2025) | 84.0 | 79.0 | 84.6 | 56.2 | 87.4 | 81.6 | 85.3 |
| | CONFSLEEPNET- | 87.4 | 82.5 | 89.0 | 58.7 | 91.4 | 84.6 | 88.6 |
| | CONFSLEEPNET | **88.9** | **84.2** | **90.2** | 61.4 | **92.4** | **86.7** | **90.2** |
| SHHS | DEEPSLEEPNET (SUPRATAK ET AL., 2017) | 81.0 | 73.9 | 85.4 | 40.5 | 82.5 | 79.3 | 81.9 |
| | XSLEEPNET (PHAN ET AL., 2022) | 87.6 | 80.7 | 92.0 | **49.9** | 88.3 | 85.0 | 88.2 |
| | SLEEPYCO (LEE ET AL., 2024) | 87.9 | 80.7 | 92.6 | 49.2 | 88.5 | 84.5 | 88.6 |
| | FLEXIBLESLEEPNET (REN ET AL., 2025) | 87.6 | 79.6 | 92.3 | 40.0 | 88.8 | 87.0 | 89.7 |
| | TMCEK (LIANG ET AL., 2025) | 84.3 | 78.0 | 90.5 | 43.2 | 87.1 | 85.1 | 83.9 |
| | CONFSLEEPNET- | 87.4 | 79.1 | 92.1 | 41.1 | 88.0 | 84.9 | 89.2 |
| | CONFSLEEPNET | **88.2** | **81.3** | **93.4** | 44.2 | **90.0** | **89.3** | **89.7** |

flict between them, $C$, is close to 1. Such that

$$\lim_{C \to 1} u^{o \overline{\nabla} b} = \lim_{C \to 1} (1-C) u^o u^b + C \frac{2 u^o u^b}{u^o + u^b}$$
$$= \frac{2}{1 + \frac{u^o}{u^b}} u^o > u^o$$

Proposition 2 demonstrates that CMAM can increase prediction uncertainty when incorporating conflicting opinions.

Notably, the proofs only address the extreme cases (i.e., $C$ approaches 1 and 0). This is because we intend to use Propositions 1 and 2 to demonstrate that our fusion operation satisfies the two proposed design principles, and the extreme cases best illustrate these properties. For the general case (i.e., $C \in (0,1)$), the conclusions of the proofs can still be extended through continuity or monotonicity arguments.

### 3.6. Multi-View Joint Training

The evidential DNNs $\{f_v(\cdot)\}_{v=1}^4$ in ConfSleepNet are trained jointly to learn view-specific evidences $\{\mathbf{e}^v\}_{v=1}^4$ from multiple views. In conventional neural network-based classifiers, the cross-entropy loss is commonly employed. However, we need to adapt the cross-entropy loss to account for the evidence-based networks. In order to extract as much evidence for the ground-truth category as possible, the loss function for the evidential DNN $f_v(\cdot)$ can be defined as:

$$\mathcal{L}_{acc}^v(\mathbf{e}^v) = \int \left[ \sum_{i=1}^K -y_i \log(p_i^v) \right] \frac{1}{B(\mathbf{e}^v + \mathbf{1})} \prod_{j=1}^K (p_j^v)^{e_j^v} \, d\boldsymbol{p}^v$$
$$= \sum_{i=1}^K y_i \left[ \psi(S^v) - \psi(e_i^v + 1) \right] \tag{11}$$

where $\psi(\cdot)$ is the digamma function, and $y_i$ is a one-hot encoder encoding the ground-truth sleep stage of the current 30-s epoch $x_i$. Note that sleep stages $N1$, $N2$ and $N3$ are treated as a single $NREM$ stage in DNN $f_3(\cdot)$ and $f_4(\cdot)$, and thus the evidences for $N1$, $N2$ and $N3$ are extracted together.

The loss function $\mathcal{L}_{acc}(\cdot)$ encourages the evidence for the ground-truth category to approach the total evidence, thereby maximizing evidence for the correct category. However, this does not guarantee suppression of evidence from incorrect classes. Such misleading evidence for a sample may not be a problem as long as it is correctly classified by the network (i.e., the evidence for the correct sleep stage is stronger than the evidence for other class labels). However, we prefer the total evidence to shrink to zero for a sleep epoch if it cannot be correctly classified, thereby showing a high uncertainty. We achieve this by incorporating an additional term into our loss function, namely Kullback-Leibler (KL) divergence:

$$\mathcal{L}_{KL}^v(\mathbf{e}^v) = KL[D(\boldsymbol{p}^v|\tilde{\mathbf{e}}^v) \ || \ D(\boldsymbol{p}^v|\mathbf{1})]$$

$$= \log\left(\frac{\Gamma(\sum_{k=1}^{K}(\tilde{e}_k^v + 1))}{\Gamma(K)\prod_{k=1}^{K}\Gamma(\tilde{e}_k^v + 1)}\right) + \quad (12)$$

$$\sum_{k=1}^{K}\tilde{e}_k^v\left[\psi(\tilde{e}_k^v + 1) - \psi(\sum_{j=1}^{K}(\tilde{e}_j^v + 1))\right]$$

where $\Gamma(\cdot)$ is the gamma function and $\psi(\cdot)$ is the digamma function, and $D(\boldsymbol{p}^v|\mathbf{1})$ is the uniform Dirichlet distribution, and $\tilde{\mathbf{e}}^v = (\mathbf{1} - \boldsymbol{y}) \odot \mathbf{e}^v$ is the Dirichlet parameters after removal of non-misleading evidences. The loss is defined as:

$$\mathcal{L}(\mathbf{e}^v) = \sum_{v=1}^{4}\mathcal{L}_{acc}^v(\mathbf{e}^v) + \lambda_t\mathcal{L}_{KL}^v(\mathbf{e}^v) \quad (13)$$

where $\lambda_t = \min(1.0, t/10) \in [0, 1]$ is the annealing coefficient, $t$ is the index of the current training epoch. By gradually increasing the weight of KL divergence in the loss through the annealing coefficient, we allow the neural network to explore the parameter space and avoid premature convergence to the uniform distribution for the misclassified samples, which may be correctly classified in future epochs.

## 4. Experiments

### 4.1. Experimental Setups

**Sleep Datasets.** We conducted extensive experiments on four publicly available datasets, including SleepEDF-20, SleepEDF-78, the Montreal Archive of Sleep Studies (MASS), and the Sleep Heart Health Study (SHHS). Detailed descriptions of each dataset and the channels used in the experiments are provided in the Appendix A.7.

**Compared Methods.** We compared ConfSleepNet with several representative baselines on multiple public datasets.

These baselines include (1) single-view networks including DeepSleepNet (Supratak et al., 2017), TinySleepNet (Supratak & Guo, 2020), SleePyCo (Lee et al., 2024), and FlexibleSleepNet (Ren et al., 2025), (2) multi-view networks including SalientSleepNet (Jia et al., 2021), XSleepNet (Phan et al., 2022), and HMDT-Net (Wang et al., 2026), and (3) uncertainty-aware networks including TMCEK (Liang et al., 2025). Detailed descriptions of all baselines are provided in Appendix A.8.

**Evaluation Metrics and Implementation Details.** To evaluate the performance of the proposed ConfSleepNet, we used accuracy (Acc) and macro-averaged F1-score (MF1). The performance on each sleep stage was evaluated using the per-class F1-score. Details of the implementation and evaluation metrics are provided in the Appendix A.9.

### 4.2. Experimental Results

**Performance Comparison.** Table 1 presents a comprehensive performance comparison between the proposed ConfSleepNet and other competing baselines. The results demonstrate that ConfSleepNet achieves the best performance among all compared methods, validating the feasibility and superiority of the proposed model. Notably, compared with representative works DeepSleepNet and XSleepNet, ConfSleepNet achieves significant improvements in classification accuracy, with gains of 2.7%-5.2% and 0.6%-2.0%, respectively. We argue that even a 1% improvement in accuracy corresponds to dozens of correctly classified samples. Therefore, ConfSleepNet has practical value in the diagnosis of sleep disorders. Furthermore, in practice, the $REM$ and $W$ stages are prone to confusion in classification due to their inherent feature similarities. Nevertheless, ConfSleepNet further improves the class-specific F1 scores for both stages. Particularly for the $REM$ stage, which holds greater clinical relevance, ConfSleepNet achieves the highest F1 scores across all datasets. Thus, we anticipate that ConfSleepNet has clinical significance for the assessment and diagnosis of sleep disorders.

We also developed a baseline model (denoted as ConfSleepNet-) based on an average-based multi-view fusion strategy and compared it with ConfSleepNet and other competing baselines. The results show that ConfSleepNet- achieves competitive performance compared to existing works, validating the effectiveness of the proposed evidential DNNs. Moreover, ConfSleepNet outperforms ConfSleepNet- by 0.8%-1.5% in accuracy. This performance gap indicates that the proposed conflict-aware multi-view aggregation method enhances model robustness against misleading predictions, thereby improving overall performance. Particularly, we compared ConfSleepNet with TMCEK (Liang et al., 2025), the current state-of-the-art uncertainty-aware baseline. The results show that Conf-

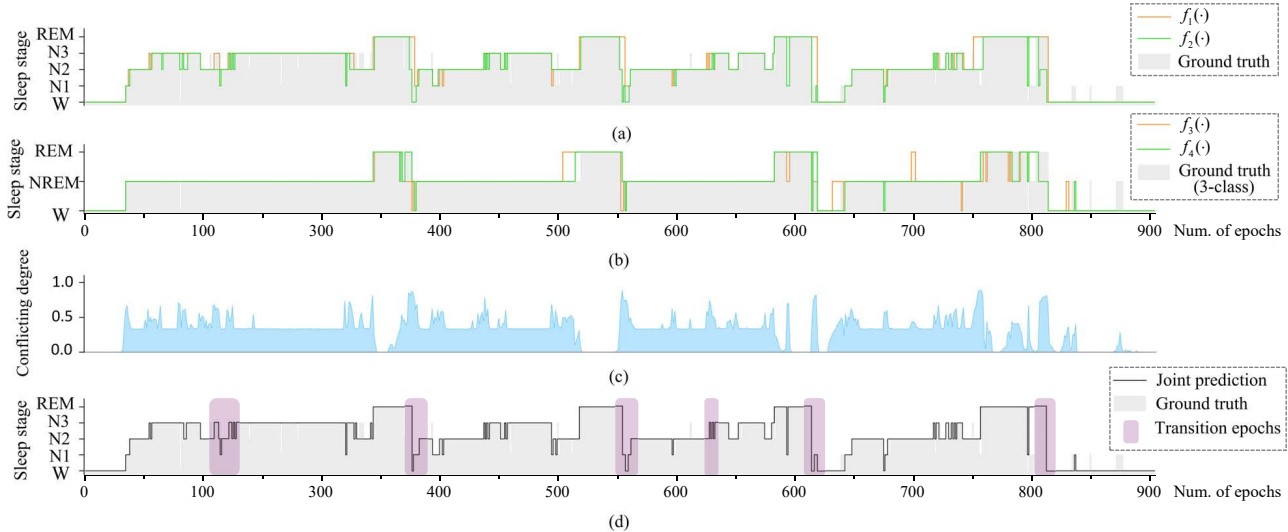

*Figure 2.* Intermediate prediction results produced by view-specific DNNs $\{f_v(\cdot)\}_{v=1}^4$ and the average degree of conflict between different views. (a) The results produced by DNN $f_1(\cdot)$ and $f_2(\cdot)$ have a 5-class structure, while (b) the results produced by $f_3(\cdot)$ and $f_4(\cdot)$ have a 3-class structure. (c) The conflict degree is typically at a low level over a sequence of epochs of the same sleep stage, and increases over transition epochs. (d) We reduce the impacts of conflictive views in making final results, which are obviously more precise than the view-specific results.

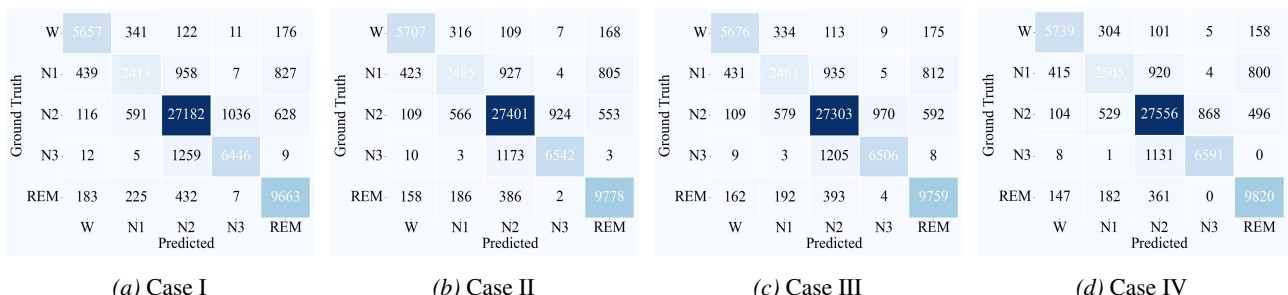

*Figure 3.* Confusion matrices on the MASS-SS3 dataset. Rows represent the true class, and columns represent the predicted class. Darker color indicates a larger number of correctly classified samples.

*Table 2.* Performance comparison between different variants of ConfSleepNet.

| METHOD | OVERALL METRICS | | F1-SCORE FOR EACH CLASS | | | | |
| --- | --- | --- | --- | --- | --- | --- | --- |
| | ACC ↑ | MF1 ↑ | $W$ | $N1$ | $N2$ | $N3$ | $REM$ |
| CASE I (CONFSLEEPNET-) | 87.4 | 82.5 | 89.0 | 58.7 | 91.4 | 84.6 | 88.6 |
| CASE II | 88.4 | 83.6 | 89.8 | 60.6 | 92.0 | 86.0 | 89.6 |
| CASE III | 88.0 | 83.2 | 89.4 | 60.0 | 91.8 | 85.5 | 89.3 |
| CASE IV (CONFSLEEPNET) | **88.9** | **84.2** | **90.2** | **61.4** | **92.4** | **86.7** | **90.2** |

SleepNet achieves accuracy gains of 2.2%-4.9% over TM-CEK, further validating the superiority of CMAM.

**Case Study.** We conducted a case study on a single patient from the MASS-SS3 dataset to investigate the conflict-aware multi-view aggregation process proposed in ConfSleepNet. Fig. 2(a) and (b) illustrate the overnight sleep staging results generated by the four evidential DNN $\{f_v(\cdot)\}_{v=1}^4$ on their respective views, along with the corresponding ground truth hypnograms. Specifically, $f_1(\cdot)$ and $f_2(\cdot)$ perform 5-class

sleep staging as shown in Fig. 2(a), while $f_3(\cdot)$ and $f_4(\cdot)$ perform 3-class sleep staging as shown in Fig. 2(b). It can be intuitively observed that due to differences in the number of input views, classification granularity, and signal characteristics, the multiple opinions generated by $\{f_v(\cdot)\}_{v=1}^4$ are not always consistent.

We quantified the degree of conflict between each pair of views using the conflict metric introduced in Section 3.4. The average conflict degree among views is presented in

*Table 3.* Performance comparison of multi-view benchmarks.

| DATASET | EDL | DCCAE | ETMC | RCML | CCML | TMCEK | CMAM (OURS) |
|---------|-----|-------|------|------|------|-------|-------------|
| HW | 97.00±0.16 | 97.05±0.24 | 98.32±0.22 | 98.70±0.19 | **98.75±0.27** | 97.75±0.42 | 98.45±0.58 |
| SCENE15 | 60.60±0.13 | 64.26±0.42 | 66.87±0.29 | 71.28±0.32 | 72.60±0.87 | 71.06±0.87 | **73.01±1.09** |
| CUB | 89.51±0.24 | 85.39±1.36 | 91.05±0.63 | 93.28±2.75 | 94.58±1.30 | 90.50±2.51 | **95.00±2.32** |
| PIE | 87.99±0.56 | 81.96±1.04 | 93.82±0.82 | 93.89±2.46 | 94.56±1.83 | 95.15±2.81 | **95.74±1.80** |

Fig. 2(c). It can be seen that during continuous $W$ or $REM$ stages, the conflict level is low (even dropping to zero), indicating a high degree of consistency among views. This observation is supported by the high F1-scores achieved for these two sleep stages, as reported in Table 1. In contrast, inter-view prediction conflicts are particularly evident during sleep stage transitions (e.g., from N2 to N3). The main reason is that transition epochs often contain features of multiple sleep stages simultaneously. We further discuss this situation in the Appendix A.10.

Fig. 2(d) presents the final prediction results after view aggregation. It can be observed that, compared to individual view-specific predictions, the aggregated result more closely aligns with the ground truth hypnogram. This is because the proposed conflict-aware multi-view aggregation method can accurately identify misleading views through uncertainty estimation and conflict assessment, thereby reducing their influence in the final decision-making process.

### 4.3. Ablation Study

To further evaluate the effectiveness of each component, we conducted ablation experiments on the MASS-SS3 dataset. Specifically, the following four model variants were constructed: (1) Case I (ConfSleepNet-): replacing the proposed CMAM with an average-based multi-view fusion strategy; (2) Case II: use the multi-view fusion method proposed by Xu et al. (2024) to replace CMAM; (3) Case III: removing the evidence mapping layer and $f_3(\cdot)$ from the evidence DNN, and performing five-class evidence learning at a single granularity for all views; (4) Case IV (ConfSleepNet): the complete version of the proposed method.

The experimental results are presented in Table 2, where Case IV achieves the best performance. Compared with Case I, the proposed CMAM significantly improves overall performance and reduces the impact of misleading views on the final predictions. The effectiveness of CMAM is further validated by comparing Case II and Case IV. In comparison with Case III, the hybrid category structure provides unique and complementary information for distinguishing sleep stages. Furthermore, Fig. 3 shows the confusion matrices of the four cases. Compared with the other ablation variants, ConfSleepNet exhibits a relative increase in correctly classified samples and a relative decrease in misclassified samples.

### 4.4. Multi-View Benchmark

To further validate the effectiveness of the proposed CMAM, we conducted experiments on four multi-view datasets: HandWritten (HW), Scene15, CUB, and PIE. The baseline methods compared include EDL (Sensoy et al., 2018), DCCAE (Wang et al., 2015), ETMC (Han et al., 2022), RCML (Xu et al., 2024), CCML (Liu et al., 2024), and TMCEK (Liang et al., 2025) (See Appendix A.11 for details). As shown in Table 3, CMAM achieves the best accuracy across multiple datasets. This strongly demonstrates that CMAM enhances model performance by quantifying the degree of conflict among views and reasonably handling the influence of such conflicts on final decision-making.

## 5. Conclusion

This work proposes ConfSleepNet, a sleep staging method based on evidential deep learning. The model utilizes parallel evidence networks to extract hybrid-granularity evidence from multi-view inputs, aiming to capture both unique and complementary information from multi-modal signals. Furthermore, this paper introduces a novel conflict-aware view aggregation strategy to fuse view-specific opinions. This method dynamically detects conflicts among views through uncertainty estimation, thereby enhancing the robustness of the final predictions. Both theoretical analysis and experimental results validate the effectiveness of ConfSleepNet in sleep stage classification.

## Acknowledgements

This work was supported by Grants 2025JC-YBQN-841 (Natural Science Foundation of Shaanxi Province), 25YJCZH244 and 24XJCZH024 (Ministry of Education of China).

## Impact Statement

Theoretical and experimental results demonstrate that the proposed method enhances both classification performance and interpretability in sleep staging tasks, which holds positive implications for promoting its practical clinical application. Experiments on multi-view tasks further validate the generalizability of the method, and future work will investigate its value in scenarios such as medical imaging.

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

# A. Appendix

In the supplemental material:

## A.1. Complementarity of Multimodal Sleep Signals

As shown in Fig. 4, EEG is commonly used for automated sleep staging tasks, with its distinctive features including theta waves (4–8 Hz) during the $N1$ sleep stage, sleep spindles and K-complex waves (12–14 Hz) in the $N2$ stage, and delta waves (0.5–4 Hz) in the $N3$ stage. Furthermore, as a modality that complements EEG-based methods, EOG exhibits significant differences during the $REM$ stage due to the characteristics of eye movements.

## A.2. Process of Feature Extraction

**High-Level Feature Extraction.** Feature extraction aims to transform raw inputs into high-level representations. As shown in Fig. 5, ConfSleepNet first employs two parallel branches with different kernel sizes to capture both temporal and frequency information from the raw input $x$. Subsequently, a global context attention module called GCNet-1D is used to further enhance the features into $h^l$ and $h^s$. It is worth noting that GCNet was originally designed to capture global understanding of visual scenes and has demonstrated superior performance (Cao et al., 2019). In this work, we modify it to adapt to the processing of one-dimensional signals. The process of basic feature extraction can be formalized as:

$$h^s = GCN^s(CNN^s(x)) \tag{14}$$

$$h^l = GCN^l(CNN^l(x)) \tag{15}$$

where $CNN$ and $GCN$ denote the processing through convolutional operations and the GCNet-1D module, respectively, and the input $x$ is either EEG or EOG.

After basic feature extraction, we employ a cross-attention mechanism to dynamically learn latent information from the interactions between the enhanced features $h^l$ and $h^s$, generating features denoted as $h^{sl}$ and $h^{ls}$. In addition, two shortcut connections are used to enable residual transmission of the enhanced features $h^l$ and $h^s$. Finally, these features are combined into a unified representation by $F = h^s \| h^l \| h^{sl} \| h^{ls}$, where $\|$ denotes feature concatenation.

**Modeling Stage-Transition Patterns.** According to the AASM guideline (Berry et al., 2012), sleep stage classification requires contextual information from neighboring epochs. Therefore, the model should be capable of learning temporal dependencies. However, the aforementioned high-level feature extraction module focuses on intra-epoch information and neglects inter-epoch stage-transition patterns. This is because convolutional neural networks are not advantageous for modeling inter-epoch dependencies. Consequently, a Bi-LSTM layer is incorporated into ConfSleepNet to model the transition patterns across a sequence of sleep epochs.

*Table 4.* Comparison of different evidence mapping strategies on the MASS-SS3 dataset.

| MAPPING STRATEGY | ACC | MF1 |
|---|---|---|
| LEARNABLE MAPPING | 88.7 | 84.1 |
| DATA-DRIVEN MAPPING | 89.1 | 84.5 |
| UNIFORM MAPPING | 88.9 | 84.2 |

### A.3. Network Architecture of the Evidential DNN

We use two types of networks to handle different inputs. For single-view EEG and single-view EOG inputs, as shown in the Fig. 6 (a), $f_1(\cdot)$ and $f_4(\cdot)$ first extract high-level representations from the raw input data, and then utilize a Bi-LSTM layer to capture stage-transition patterns between epochs. The multi-view network is designed for multi-view inputs, and its difference from the single-view network lies in the use of two independent feature extractors. As illustrated in the Fig. 6 (b), two parallel branches are used to process EEG and EOG inputs, respectively, and fusion is performed after feature extraction.

After obtaining the high-level representation of the sleep input, different from deep learning methods, we use an output layer to generate non-negative evidence vectors. Specifically, a fully connected layer first transforms the high-dimensional features into evidence vectors of fixed size (typically the total number of classes). Then, the *SoftPlus* activation function is applied to ensure the non-negativity of the evidence.

### A.4. Feasibility of the Mapping Strategy.

We adopt a uniform mapping strategy to convert coarse-grained evidence into fine-grained evidence. To validate the feasibility of this mapping strategy, as shown in the Table 4, we developed two variants and conducted comparative experiments on the MASS-SS3 dataset. The two variants are learnable mapping and data-driven mapping.

The experimental results show that the performance differences among the three mapping strategies are minimal, indicating that the uniform assignment hypothesis does not introduce significant bias in the current dataset. This can be attributed to the following two reasons. First, the evidence mapping is performed after the training of the coarse-grained DNNs, where the $NREM$ evidence has already encoded the characteristics of the entire $NREM$ category. Second, the subsequent conflict-aware aggregation and multi-view joint training compensate, to some extent, for the simplification introduced by the mapping layer. Ultimately, considering the trade-off between model performance and training time, we select the uniform mapping as the evidence mapping strategy.

### A.5. Proof of Subjective Logic Constraints

*Proof.*

$$
\sum_k b_k^{a\bar{\triangledown}b} + u^{a\bar{\triangledown}b}
$$

$$
= \frac{u^a(1-u^b) + u^b(1-u^a) + (1-C)u^a u^b(2-u^a-u^b)}{u^a + u^b}
$$

$$
+ \frac{(1-C)u^a u^b(u^a+u^b) + 2Cu^a u^b}{u^a + u^b}
$$

$$
= \frac{u^a + u^b - 2u^a u^b + 2(1-C)u^a u^b + 2Cu^a u^b}{u^a + u^b}
$$

$$
= 1
$$

### A.6. Limitations of the Proposed CMAM

Consistent with existing EDL-based multi-view aggregation methods (Han et al., 2022; Xu et al., 2024), CMAM suffers from performance limitations due to its failure to satisfy the commutativity law (e.g., $(\mathcal{M}^1\bar{\triangledown}\mathcal{M}^2)\bar{\triangledown}\mathcal{M}^3 \neq \mathcal{M}^1(\bar{\triangledown}\mathcal{M}^2\bar{\triangledown}\mathcal{M}^3)$). Specifically, the degree of conflict $C$ depends on the specific pair of opinions being combined. When fusing opinions generated from three or more views, the initial fusion (i.e., $\mathcal{M}^1\bar{\triangledown}\mathcal{M}^2$) alters the original belief mass distribution, which in turn affects the degree of conflict with $\mathcal{M}^3$. Although the associativity law does not hold, the definition in Eq. (10) remains

*Table 5.* A summary of datasets.

| DATASET | # SUBJECTS | # RECORDINGS | EEG CHANNEL | EOG CHANNEL | SCORING RULE | SAMPLING RATE |
|---|---|---|---|---|---|---|
| SLEEPEDF-20 | 20 | 39 | FPZ-CZ | ROC-LOC | R&K | 100Hz |
| SLEEPEDF-78 | 78 | 153 | FPZ-CZ | ROC-LOC | R&K | 100Hz |
| MASS-SS3 | 62 | 62 | C4-LER | EOG LEFT HORIZ | AASM | 128Hz |
| SHHS | 329 | 329 | C4-A1 | ROC-LOC | R&K | 125Hz |

valid because we adopt a fixed order for pairwise fusion, thereby ensuring a deterministic result. In summary, this limitation will be thoroughly investigated in future work.

### A.7. Datasets

We used four public datasets, including SleepEDF-20[1], SleepEDF-78[2], MASS-SS3[3], and a large-scale dataset, SHHS[4]. A summary of the datasets is shown in Table 5.

**SleepEDF-20.** This dataset includes 20 healthy participants (10 males and 10 females) aging from 25 to 34 years (Kemp et al., 2000) and manually labeled according to the R&K manual. For each subject, two consecutive day-night PSG recordings are collected, except for subject 13 who has one night's data lost due to device failure. This dataset contains two EEG channels (Fpz-Cz and Pz-Oz) and a horizontal EOG channel. The sampling rate for all EEG and EOG signals is 100 Hz. Following previous studies (Phan et al., 2022; 2019; Phyo et al., 2023), we utilize the Fpz-Cz EEG and EOG channels, and merge $N3$ and $N4$ into a single $N3$ stage based on the AASM rule.

**SleepEDF-78.** As an expanded version of SleepEDF-20 (Goldberger et al., 2000), SleepEDF-78 contains a total of 78 subjects ranging in age from 25 to 101 years, and comprises 153 whole-night PSG sleep recordings. The other settings are the same as SleepEDF-20.

**MASS-SS3.** This dataset is composed of 62 nights from healthy subjects. Each recording contains 20 EEG channels and 2 EOG channels. Manual annotation is performed by sleep experts according to the AASM standard. All EEG and EOG signals have a sampling rate of 256 Hz. We employ the C4-LER EEG channel and the Left Horiz EOG channel, and downsample the signals to 128 Hz.

**SHHS.** The SHHS dataset (Quan et al., 1997) is a large-scale, multi-center dataset designed to investigate the association between sleep-disordered breathing and cardiovascular disease. Since the participants in this dataset were drawn from multiple existing epidemiological cohort studies, to reduce the influence of various disease-related factors, we followed the subject selection criteria proposed by Fonseca et al. (2016). and selected 329 individuals with relatively healthy sleep patterns (i.e., an Apnea-Hypopnea Index below 5). For this filtered subset, we employed the C4-A1 EEG channel with a sampling rate of 125 Hz, as well as the EOG channel, in our experiments.

### A.8. Compared Methods

We compared our model with the following eight baselines:

- **DeepSleepNet** (Supratak et al., 2017) is an automatic sleep staging model based on single-channel EEG. It constructs a convolutional neural network (CNN) to extract time-domain features from raw signals and employs a Bi-LSTM network to automatically learn transition rules between sleep stages from EEG epochs.

- **TinySleepNet** (Supratak & Guo, 2020) is an end-to-end model capable of performing automatic sleep staging with relatively limited training data and computational resources. Through data augmentation, the model becomes more robust to temporal shifts and avoids overfitting to the sequence order of sleep stages.

---

[1] https://www.physionet.org/content/sleep-edf/1.0.0/
[2] https://www.physionet.org/content/sleep-edf/1.0.0/
[3] https://www.ceams-carsm.ca/en/MASS
[4] https://www.sleepdata.org/datasets/shhs

- **SalientSleepNet** (Jia et al., 2021) is a temporal fully convolutional network based on the U2-Net architecture (Qin et al., 2020), consisting of two independent U2-shaped streams that extract salient features from multimodal data. It's designed with multi-scale extraction and multimodal attention modules help the model achieve excellent performance.

- **XSleepNet** (Phan et al., 2022) is a sequence-to-sequence sleep staging model capable of learning joint representations from raw signals and time-frequency images. Its characteristic lies in preserving the representational capacity of different views while enhancing robustness to training data.

- **SleePyCo** (Lee et al., 2024) employs a feature pyramid backbone network to extract multi-scale temporal and frequency features from raw EEG signals. Furthermore, through supervised contrastive learning, the method reduces the distance between features of the same class and increases the distance between features of different classes, demonstrating particularly strong discriminative ability for the $N1$ and $REM$ sleep stages.

- **FlexibleSleepNet** (Ren et al., 2025) is a lightweight classification model based on adaptive feature extraction (AFE) and scaling variation compression (SVC). AFE and SVC compress and expand the dimensions of features captured from multi-channel data, enabling the network to effectively learn spatiotemporal dependencies across channels.

- **TMCEK** (Liang et al., 2025) is a trusted multi-view classification method that has achieved notable results in automatic sleep staging tasks. It integrates expert knowledge to improve feature interpretability and introduces a distribution-aware subjective opinion mechanism for more reliable confidence estimation.

- **HMDT-Net** (Wang et al., 2026) aims to mitigate the impact of inter-subject variability on sleep staging models. The method proposes a trusted fusion strategy to effectively integrate heterogeneous multimodal data, while incorporating adversarial learning to extract common feature representations across subjects.

### A.9. Implementation Details and Evaluation Metrics

**Implementation Details.** The proposed ConfSleepNet model and its variants were implemented using the PyTorch framework (version 1.9), and trained on an Nvidia GeForce RTX 3090 GPU with a total number of 100 training epochs. The models were trained with a batch size of 16, and the Adam optimizer (Kingma & Ba, 2014) is employed with a learning rate of 1e-3 for the loss function defined in Eq. (13). The multi-view input signals were segmented into sequences of 20 epochs (i.e., each sequence is composed of 20 sleep epochs). We evaluated our proposed model by using a k-fold cross-validation scheme, where k was set to 20, 10, 31, and 5 for SleepEDF-20, SleepEDF-78, MASS-SS3, and SHHS datasets, respectively. In each fold, we selected one group of subjects as testing data, one group for validation, and the remaining groups for training. For example, subject-wise 20-fold cross-validation on SleepEDF-20 with 20 subjects was a leave-one-subject-out cross-validation.

**Evaluation Metrics.** We employed two performance metrics, accuracy (Acc) and macro-averaged F1-score (MF1), to evaluate the model. They are defined as follows:

$$Acc = \frac{TP + TN}{TP + TN + FP + FN} \tag{16}$$

$$MF1 = \frac{\sum_{i=1}^{K} F1_i}{K} \tag{17}$$

where $TP$ is the number of true positives, $TN$ is the number of true negatives, $FP$ is the number of false positives, and $FN$ is the number of false negatives. $F1 = \frac{2 \times Pre \times Rex}{Pre + Rec}$, $Rec = \frac{TP}{TP+FN}$, $Pre = \frac{TP}{TP+FP}$, and $K$ denotes the number of sleep stage classes. We further computed per-class metrics by considering a single class as a positive class and all other classes combined as a negative class.

### A.10. Conflicts Among View-Specific Opinions

We calculated the sample distribution of a subject under different conflict degrees, as well as the proportions of different conflict intervals during sleep transition and non-transition periods. The results are presented in Tables 6 and 7. Statistical results show that nearly half (43.27%) of high-conflict instances occur during the transition period, a proportion significantly higher than that of other conflict levels, indicating a strong correlation between high conflict and the transition period.

*Table 6.* Sample distribution of a subject in MASS-SS3 under different conflict degrees.

| TOTAL EPOCHS | LOW CONFLICT (0-0.3) | MIDDLE CONFLICT (0.3-0.6) | HIGH CONFLICT (0.6-1) | AVERAGE |
|---|---|---|---|---|
| 774 | 218 (28.17%) | 407 (52.58%) | 149 (19.25%) | 0.431 |

*Table 7.* Proportions of different conflict intervals in transition and non-transition periods

| | HIGH (0.6-1) | MIDDLE (0.3-0.6) | LOW (0.0-0.3) |
|---|---|---|---|
| PROPORTION IN TRANSITION PERIOD | 43.28% | 8.85% | 7.34% |
| PROPORTION OF NON-TRANSITION PERIOD | 56.72% | 91.15% | 92.66% |
| RATIO OF TRANSITION PERIOD | – | 4.89 | 5.90 |

The main reasons can be summarized as follows: (1) At the signal feature level, the transition between adjacent sleep stages involves mixed physiological features, to which EEG and EOG have different sensitivities; (2) The model inherently possesses uncertainty when modeling the boundaries of sleep stages; (3) Accurately identifying the transition period is also challenging for human experts, meaning that subjective judgment discrepancies most easily occur in this context, reflecting the inherent difficulty of the sleep stage classification task, a viewpoint supported by existing related works (Chen et al., 2025).

### A.11. Multi-View Benchmark

**Multi-View Dataset.** We used four multi-view datasets: HandWritten[5], Scene15[6], CUB[7], and PIE[8]. Similar to previous work (Han et al., 2022), we extracted multi-view features from different datasets, with detailed descriptions of each dataset provided below:

- **HandWritten** dataset consists of handwritten numerals (0–9) extracted from Dutch utility maps, with 200 instances per class (2,000 samples in total), where these numerals were converted into binary images and characterized using six feature sets.

- **Scene15** dataset was designed specifically for image scene classification tasks and contains 4,485 images spanning 15 common indoor and outdoor scene categories, serving as a widely adopted benchmark for comparing various algorithms.

- **CUB** dataset is the most widely used dataset for fine-grained visual classification tasks, comprising 11,788 images belonging to 200 bird subcategories.

- **PIE** dataset contains 680 facial images from 68 subjects.

**Baseline Methods.** We used six competitive baselines to evaluate CMAM:

- **EDL** (Sensoy et al., 2018) explicitly models the predictive uncertainty of a model using subjective logic theory. Specifically, it imposes a Dirichlet distribution over class probabilities, thereby transforming the neural network's predictions into subjective opinions. This model has achieved notable success in out-of-distribution sample detection and exhibits strong robustness to interference.

- **DCCAE** (Wang et al., 2015) builds upon deep canonical correlation analysis (DCCA) by introducing an autoencoder's reconstruction loss as a regularization term, effectively combining both objectives. This approach performs prominently in tasks that require both cross-view prediction capability and the preservation of single-view information.

- **ETMC** (Han et al., 2022) is a trustworthy multi-view classification method that dynamically integrates different views at the evidence level to enhance classification reliability. This framework accurately identifies predictive uncertainty while endowing the model with robustness against potential noise.

---

[5] https://archive.ics.uci.edu/dataset/72/multiple+features
[6] https://figshare.com/articles/dataset/15-Scene_Image_Dataset/7007177/1
[7] https://www.vision.caltech.edu/visipedia/CUB-200.html
[8] http://www.cs.cmu.edu/afs/cs/project/PIE/MultiPie/Home.html

- **RCML** (Xu et al., 2024) addresses a novel problem of reliable conflict multi-view learning, which requires models to provide trustworthy decision results for conflicting multi-view data. To this end, it develops an evidential conflict multi-view learning approach that aggregates view-specific opinions through a conflicting opinion aggregation strategy.

- **CCML** (Liu et al., 2024) proposes a consistency and complementarity-aware multi-view learning method. It dynamically decouples consistent and complementary evidence, which is then processed using corresponding principles.

- **TMCEK** (Liang et al., 2025) integrates expert knowledge with a distribution-aware subjective opinion mechanism, thereby enhancing feature interpretability while achieving more reliable confidence estimation.

**Uncertainty Evaluation.** To further evaluate the advantage of CMAM in uncertainty awareness, we visualized the data density distributions under different noise levels ($\sigma$) on two multi-view datasets, as illustrated in the Fig 7. The experimental results show that, compared with the clean datasets, the density of high-uncertainty distributions is more prominent on the noisy datasets. This demonstrates that CMAM can effectively identify noisy instances and estimate their uncertainty.

### A.12. Supplementary Figures

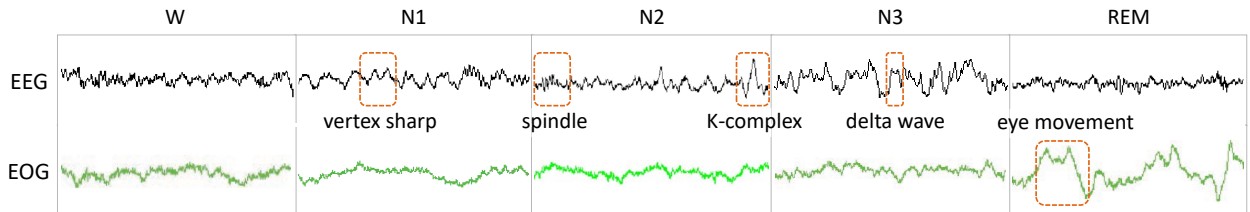

*Figure 4.* Stage-related features in EEG and EOG signals.

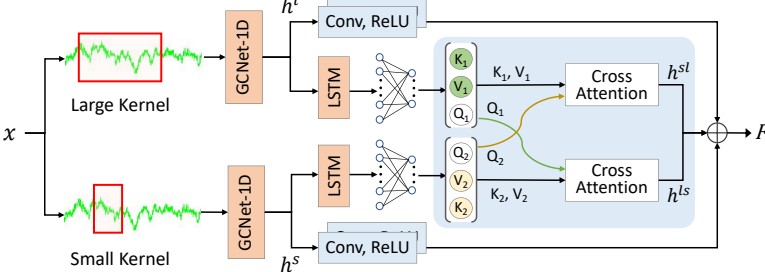

*Figure 5.* Architecture of feature extraction that utilizes two branches with varying kernel sizes and a cross-attention mechanism.

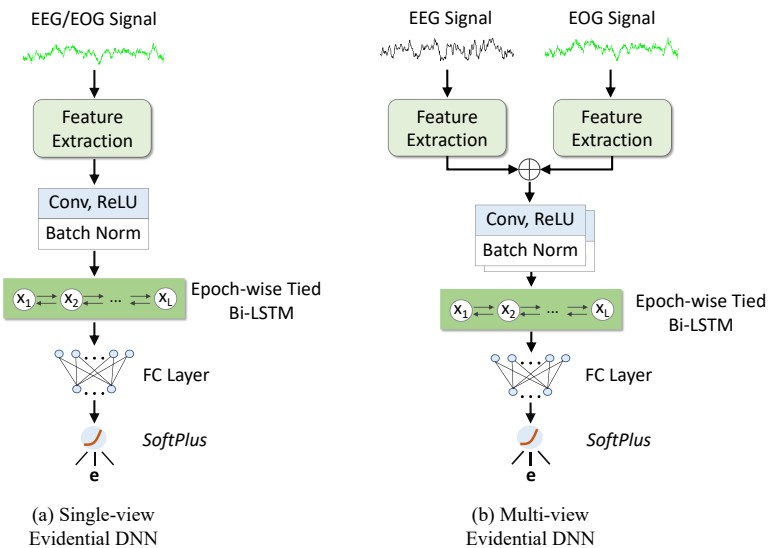

*Figure 6.* The Network Architecture of the Evidential DNN.

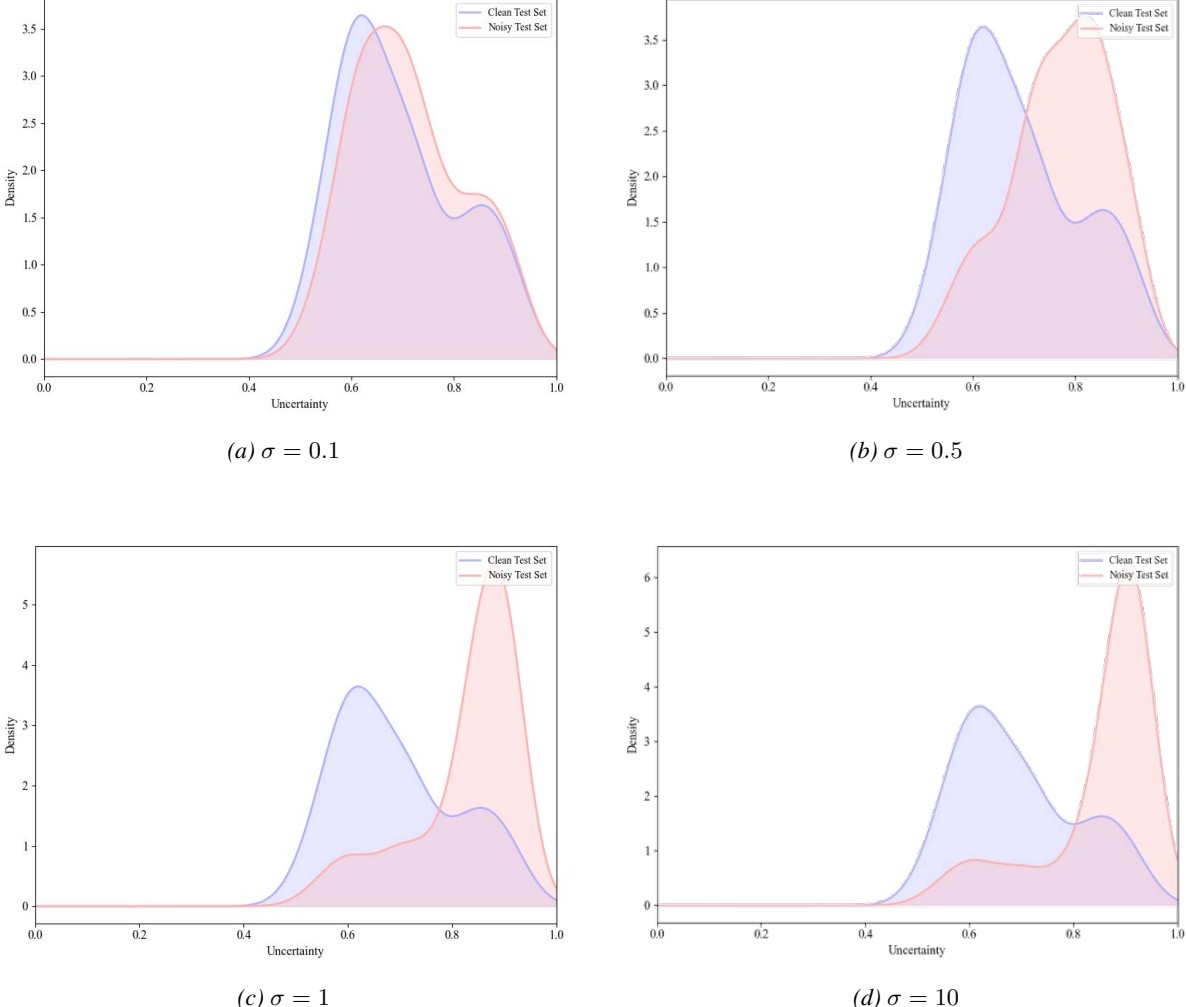

*Figure 7.* Density of uncertainty on the PIE dataset. As the noise intensity increases, the uncertainty curves of conflicting instances also increase.

