# OpenReview forum: "A Conflict-aware Evidential Framework for Reliable Sleep Stage Classification"
_ICML.cc/2026/Conference — ICML 2026 regular_

### Official Review · Reviewer_KEeD · 2026-03-04

**Soundness:** 2
**Presentation:** 3
**Significance:** 3
**Originality:** 3
**Overall Recommendation:** 3
**Confidence:** 4

**Summary:**

This paper proposes ConfSleepNet, a conflict-aware evidential learning framework for multiview sleep staging that targets real-world cases where modalities are misaligned and may conflict. It first learns per-view class evidence with multi-branch evidential DNNs using a hybrid label granularity, and then performs fusion by modeling each view as a Dirichlet opinion. The proposed CMAM explicitly measures inter-view conflict and adjusts uncertainty accordingly—reducing it when views agree and increasing it under conflicting, uncertain evidence—to yield a more reliable joint decision. Experiments on SleepEDF-20, SleepEDF-78, and MASS-SS3 report improvements in Accuracy and Macro-F1 over several baselines, supported by ablations and case studies on transition-related conflicts.

**Compliance With Llm Reviewing Policy:**

Affirmed.

**Key Questions For Authors:**

nan

**Limitations:**

yes

**Strengths And Weaknesses:**

Strengths

1. Realistic problem setting: The paper clearly highlights that multiview sleep staging in practice faces modality misalignment and conflicts, where naïve concatenation or fixed-weight fusion can be unreliable.

2. Physiologically motivated hybrid labeling: Motivated by clinical priors (EEG better at fine-grained NREM staging; EOG better at REM), the hybrid 3-class (W/REM/NREM) + 5-class design, plus a mapping layer to unify predictions, is conceptually clear and well-motivated.

Weaknesses

1. Limited dataset diversity: Although results are reported on three datasets (SleepEDF-20, SleepEDF-78, MASS-SS3), SleepEDF-20 and SleepEDF-78 are essentially two splits/subsets of the same source dataset, so the evaluation is closer to two independent data sources (SleepEDF + MASS), limiting claims about broad cross-domain generalization.

2. Baseline suite is small and dated: Compared to the paper’s focus on conflict/uncertainty-robust fusion, the baseline coverage is limited and includes several older or conventional sleep-staging models; stronger and more recent multiview dynamic-weighting, uncertainty-aware fusion, and conflict-modeling baselines are missing.

3. Gains are modest and class-wise superiority is inconsistent: The Macro-F1 improvements are not consistently large, and per-class F1 is often not the best for ConfSleepNet, which weakens the claim of broadly improved reliability across stages.

4. Potentially strong “equal-split” assumption in the mapping layer: The evidence mapping distributes NREM evidence uniformly to N1/N2/N3 based on the assumption that these subclasses share the same distribution when learning NREM, which is often unrealistic given real-world class imbalance and distinctive stage morphology; this may introduce systematic bias, especially in imbalanced or transitional segments. A learnable/adaptive mapping baseline would strengthen the argument. In addition, comparisons against low-quality-modality-robust fusion and uncertainty-aware sleep-staging baselines are insufficient.

5. Insufficient quantitative analysis on transitions: While the paper argues conflicts concentrate around stage transitions and provides qualitative examples, it lacks stratified quantitative metrics (transition vs non-transition) to substantiate the central narrative.

6. Reliability is evaluated mainly via standard accuracy metrics: Despite emphasizing “reliable decision-making” and better uncertainty handling, experiments primarily report Accuracy/Macro-F1/per-class F1; uncertainty-centric reliability metrics (e.g., calibration error, selective risk, abstention/alert curves, or uncertainty behavior under noise/OOD) are missing, making it difficult to confirm improved reliability rather than just improved accuracy.

---

> ### Author Rebuttal · Authors · 2026-03-31
>
> We sincerely thank you for your constructive suggestions, which are crucial for improving the quality of our paper. Based on your comments, our point-by-point responses are as follows.
>
> ---
> **A1:** We agree that relying solely on SleepEDF‑20 and SleepEDF78 is insufficient for demonstrating cross‑domain generalization. To address this, we conducted additional experiments on the SHHS dataset. The results (dataset description and full results available at: https://anonymous.4open.science/r/KEeD-2325) show that ConfSleepNet still achieves the best performance (Acc = 88.2%), which strongly supports its cross‑domain generalization ability. We will add this experiment and corresponding analysis in the revised manuscript.
>
> **A2:** Based on the valuable comment, we have made corresponding improvements of the experiments:
>
> - We have added the latest baseline methods on all datasets, including SleePyCo (ESWA 2024), FlexibleSleepNet (JBHI 2025), and HMDT-Net (IEEE TETCI 2026).
>
> - We compared our proposed conflict-aware multi-view aggregation method (CMAM) with representative multi-view learning works (covering dynamic weighting, uncertainty-aware fusion, and conflict modeling) on multi-view datasets. Simplified experimental results (full results are available at https://anonymous.4open.science/r/KEeD-2325) are as follows:
>
> |Dataset|RCML|TMCEK|CMAM|
> |-|-|-|-|
> |HandWritten|98.70±0.19|97.75±0.42|98.45±0.58|
> |PIE|93.89±2.46|95.15±2.81|95.74±1.80|
>
> We will supplement the datasets, baselines, and experimental setup in the revision.
>
> **A3:** We acknowledge that ConfSleepNet does not achieve the highest per-class F1 scores across all sleep stages. In fact, ConfSleepNet achieves the best Acc and MF1 on all datasets, demonstrating the advantage of this method in overall performance. We also note that on most sleep stages, especially clinically important stages such as the N3 stage, ConfSleepNet achieves the best performance, which is crucial for real-world clinical applications dealing with diverse sleep patterns.
>
> **A4:** Thanks for your suggestion. We acknowledge the limitations of the uniform NREM evidence mapping assumption and the insufficient baseline comparisons. Based on this comment, we developed two variants (e.g., learnable mapping and data-driven mapping strategy) and conducted additional experiments. Results are as follows:
>
> |Mapping Strategy|Acc|MF1|
> |-|-|-|
> |Learnable Mapping|88.7|84.1|
> |Data-driven Mapping|89.1|84.5|
> |Uniform Mapping|88.9|84.2|
>
> The experimental results show that the performance differences among the three mapping strategies are minimal, indicating that the uniform assignment assumption does not introduce significant bias in the current datasets. This may be because:
>
> 1) Evidence mapping occurs after coarse-grained DNN training, where the NREM evidence has already encoded the overall NREM characteristics.
>
> 2) The subsequent conflict-aware fusion and multi-view joint training compensate for the simplification of the mapping layer.
>
> Nevertheless, we agree with your point that learnable mapping is theoretically superior, and we will supplement the experiments and analysis of this variant in the revision.
>
> - We compared against two additional baselines on multiple datasets: TrustSleepNet (uncertainty-aware sleep staging) and XSleepFusion (low-quality modality robust fusion). Experimental results are available at https://anonymous.4open.science/r/KEeD-2325.
>
> **A5:** We analyzed the conflict values between different view opinions for a specific subject in the MASS-SS3 dataset. The number and proportion of epochs in each conflict interval, along with the average conflict values, are as follows:
>
> |Epoch|Low (0.0-0.3)|Middle (0.3-0.6)|High (0.6-1)|Avg. Conflict|
> |-|-|-|-|-|
> |774|218 (28.17%)| 407 (52.58%)|149 (19.25%)|0.431|
>
> We calculated the proportions of different conflict intervals during transition and non-transition periods as follows:
>
> |Interval|Proportion in transition period|Proportion of non-transition period|Ratio of transition period|
> |-|-|-|-|
> |High (0.6-1)|43.28%|56.72%|/|
> |Middle (0.3-0.6)|8.85%|91.15%|4.89|
> |Low (0.0-0.3)|7.34%|92.66%|5.90|
>
> The results show that nearly half of the high-conflict epochs occur during transition periods, with a proportion significantly higher than that of other conflict intervals, demonstrating a strong correlation between high conflict and transition epochs.
>
> **A6:** We acknowledge that the current metrics are insufficient to demonstrate "reliable decision-making". To further evaluate uncertainty, we visualized the density distributions of clean and noisy datasets under different noise levels across multiple datasets. Please refer to: https://anonymous.4open.science/r/KEeD-2325. The results demonstrate that our method can effectively identify noisy instances, validating its capability in uncertainty estimation.
>
> ---
> We sincerely thank you again for your valuable suggestions. We will further improve the quality of our work in the revision.

---

### Official Review · Reviewer_r2CP · 2026-03-09

**Soundness:** 3
**Presentation:** 3
**Significance:** 3
**Originality:** 3
**Overall Recommendation:** 4
**Confidence:** 4

**Summary:**

This paper proposes a multi-view sleep stage classification framework. The proposed method comprises two parts: (i) evidential DNNs that extract evidence from EEG and EOG inputs, and (ii) conflict-aware opinion aggregation.

There are four evidental DNNs: (i) a DNN that maps EEG into a fine-grained evidence, (ii) a DNN that maps EOG into a coarse-grained evidence, (iii) a DNN that maps EEG and EOG into a fine-grained evidence, and (iv) a DNN that maps EEG and EOG into a coarse-grained evidence.

The coarse-grained evidences are mapped to fine-grained evidence by the matrix $U$.

The evidence of each view is used as parameter of a view-specific Dirichlet distribution. The view-specific opinions are aggregated in a conflict-aware manner.

**Compliance With Llm Reviewing Policy:**

Affirmed.

**Final Justification:**

The authors have clearly addressed my concerns, and I have increased my score accordingly (3 -> 4).

**Key Questions For Authors:**

Q1. Do the commutative and associative laws hold for Eq. (7)? If so, proof is required. If not, Eq. (10) is not well-defined.

Q2. Authors state that proof of proposition 1 and 2 are provided in A.5 and A.6 respectively. However, A.5 proves that uncertainty decreases when $C=0$, and A.6 proves that uncertainty increases when $C=1$. Therefore, the proofs only support the case when opinions are either perfectly consistent or conflictive.

Q3. In page 2, line 055, why designing special subnetworks cannot detect noisy or conflictive views?

Q4. Figure 1 do not match the main text. Each output of each evidential DNN is used to construct view-specific opinion, however, the outputs of $f_1$ and $f_2$ are appear to be merged together before constructing opinion. Likewise, the outputs of $f_3$ and $f_4$ seems to be merged. Therefore, Fig. 1 should be changed.

Q5. There is a type-o in page 5, line 239:
"(2) $C=1$ indicates completely conflicting opinions." may be right.

**Limitations:**

Yes

**Strengths And Weaknesses:**

(i) Soundness:
The extension from Eq. (7)--(8) to the case of $n>2$ in Eq. (10) is not straightforward. Do the commutative and associative laws hold for Eq. (7)? If so, proof is required. If not, Eq. (10) is not well-defined.

Authors state that proof of proposition 1 and 2 are provided in A.5 and A.6 respectively. However, A.5 proves that uncertainty decrases when $C=0$, and A.6 proves that uncertainty increases when $C=1$. This provides the case when opinions are either perfectly consistent or conflictive.

In page 2, line 055, it is unclear why designing special subnetworks cannot detect noisy or conflictive views.

(ii) Presentation:
In the current form, there is no description what is "inter-view conflict" in the introduction part, so it is hard to understand.

Figure 1 does not match the main text. In its current form, the outputs of networks $f_1$ and $f_2$ appear to be merged together. Likewise, the outputs of networks $f_3$ and $f_4$ also seem to be merged.

(iii) Significance
This work may unlock new direction in both sleep staging and multi-view learning.

(iv) Originality
This work addresses inter-view conflict problem in sleep stage classification that was not considred previously.

---

> ### Author Rebuttal · Authors · 2026-03-29
>
> We sincerely appreciate your constructive suggestions, which are highly valuable for enhancing the quality of our paper. We have thoroughly considered all the points raised and provided our point-by-point responses below.
>
> ---
> **Response to Weaknesses**
>
> We apologize for the unclear description of “inter-view conflict” in the original introduction. We clarify that inter-view conflict is defined as discrepancies in belief mass distributions across views (EEG/EOG) on the same instance. The following are two examples with view $f_1$ and view $f_4$:
>
> - **Same predicted class:**
>
> |View|W|N1|N2|N3|REM|
> |-|-|-|-|-|-|
> |$f_1$|0.65|0.10|0.08|0.07|0.05|
> |$f_4$|0.50|0.20|0.10|0.08|0.07|
>
> Both views predict the W stage, but EEG has a belief of 0.65 while EOG has 0.50. This discrepancy reflects inter-view conflict.
>
> - **Different predicted class:**
>
> |View|W|N1|N2|N3|REM|
> |-|-|-|-|-|-|
> |$f_1$|0.70|0.05|0.08|0.07|0.05|
> |$f_4$|0.20|0.50|0.10|0.08|0.07|
>
> The difference in stages supported by EEG/EOG demonstrates conflict manifested as inconsistent opinions. Both cases arise from the differing sensitivity of EEG/EOG to sleep stage features. We will clarify this term in the revision.
>
> ---
> **A1:** Thank you for raising this insightful suggestion. The fusion operator in Eq. (7) satisfies the commutative law but not the associative law.
>
> - **Commutative law:** The opinion consists of belief masses $b_k$ and uncertainty $u$, as defined in Eqs. (8) and (9).  $C$ is defined as $C(\mathcal{M}^a, \mathcal{M}^b) = 1 - \frac{\sum_k b_k^a \cdot b_k^b}{\sum_i b_i^a \cdot \sum_j b_j^b}$. An observation is that $C(\mathcal{M}^a, \mathcal{M}^b)$ is symmetric. Substituting $C$ into Eqs. (8) and (9), all expressions are symmetric with respect to $a$ and $b$. Thus, Eq. (7) satisfies commutativity.
>
> - **Associative law:** Eq. (7) does not satisfy the associative law because $C$ depends on the specific pair of opinions. When fusing three opinions, the first fusion in $(\mathcal{M}^a \bar{\nabla} \mathcal{M}^b) \bar{\nabla} \mathcal{M}^c$ changes the belief distribution, affecting the conflict with $\mathcal{M}^c$.
>
> - **Well-definedness of Eq. (10):** Although associativity does not hold, Eq. (10) remains well-defined in practice because we adopt a fixed order for sequential pairwise fusion, ensuring a deterministic result. Existing evidential multi-view learning works (e.g., ETMC [1], RCML [2]) also face the issue of fusion order affecting performance. We will discuss this limitation and leave a deeper solution for future work.
>
> ---
> **A2:** The proofs in Appx. A.5 and A.6 only address the limiting cases of $C \to 0$ (perfectly consistent) and $C \to 1$ (completely conflicting). We clarify:
>
> - The purpose of Propositions 1 and 2 is to demonstrate two fundamental properties of our fusion operator:
>
> 1) Incorporating a consistent opinion reduces uncertainty.
>
> 2) Incorporating a conflicting opinion increases uncertainty.
>
> The limiting cases best illustrate these properties.
>
> - For general cases with $C \in (0,1)$, the conclusions can be extended via continuity or monotonicity. From Eq. (8), $u^{a \oplus b}$ is linear in $C$ with a positive coefficient (since $\frac{2}{u^a+u^b} > 1$ for $u^a, u^b \in [0,1)$). Thus, $u^{a \oplus b}$ is strictly increasing in $C$. By continuity:
>
> 1) When $C$ is small, $u^{a \oplus b} < u^a$.
>
> 2) When $C$ is large, $u^{a \oplus b} > u^a$ (given $u^b > u^a$).
>
> We will supplement this argument in the revision.
>
> ---
> **A3:** Thank you for pointing out this problem. We clarify our intention as follows:
>
> Not all specialized subnetworks fail to detect conflicts; rather, subnetworks designed with feature-level fusion (e.g., attention-based mechanisms) cannot explicitly identify or quantify conflicts. Although most existing works can adaptively learn the weights of different views, they essentially learn only view saliency or relevance, overlooking view reliability or conflicts.
>
> Attention-based subnetworks typically:
>
> 1) Learn from statistical correlations between features and labels.
> 2) Do not explicitly model conflicts. As a result, they are unable to detect noisy or conflicting views.
>
> In contrast, our method constructs view-specific opinions independently at the decision level and explicitly models disagreements between views through conflict measurement, thereby dynamically reducing the weights of conflicting views. We will add this explanation to the revision to clarify the original statement.
>
> ---
> **A4:** We apologize for our carelessness. We acknowledge that this caused confusion for readers and have therefore revised Fig.1 at: https://anonymous.4open.science/r/Rebuttal-r2CP-4BA6.
>
> ---
> **A5:** It is indeed our problem. We acknowledge this and will correct it in the revision.
>
> ---
> Thank you once again for your time and expertise.
>
> ---
> **Reference**
>
> [1] Han Z et al., Trusted multiview classification with dynamic evidential fusion. TPAMI, 2023.
>
> [2] Xu C et al., Reliable conflictive multi-view learning. AAAI, 2024.

---

> > ### Author Rebuttal · Reviewer_r2CP · 2026-04-01
> >
> > Thanks to the authors for addressing my concerns carefully and clearly.
> > I have adjusted my score according to the authors rebuttal.

---

> > > ### Author Response · Authors · 2026-04-02
> > >
> > > Thank you for your positive feedback and meticulous review. Your insightful comments are greatly appreciated, and we will make every effort to enhance our manuscript accordingly.
> > >
> > > Best wishes.

---

### Official Review · Reviewer_ugUx · 2026-03-10

**Soundness:** 3
**Presentation:** 4
**Significance:** 2
**Originality:** 3
**Overall Recommendation:** 4
**Confidence:** 3

**Summary:**

This paper studies the application of evidential deep learning for the sleep stage classification task, where  different streams of data are used to conduct classification by fusing information from each source. The key challenge in this work is how to fuse data sources with different granularities while addressing potential conflicts between them, by employing a systematic uncertainty-aware framework. They propose a conflict-aware multi-view fusion method called ConfSleepNet that, instead of a typical softmax output, parametrises a Dirichlet distribution using the so-called evidence and is trained with the "Dirichlet expected cross-entropy loss" instead of cross-entropy loss. The evidence is obtained by processing the signal in three distinct steps: 1) evidence extraction, 2) evidence mapping that reconciles any difference in the number of classes/categories between each source, and 3) forming and fusing opinions as part of the Dirichlet parameterisation. They provide propositions that support their design principles and complement them with empirical experiments to validate their approach on three different datasets compared to state-of-the-art models, accompanied by an ablation study. Empirical evidence suggests improvement over the baselines, albeit marginal.

**Compliance With Llm Reviewing Policy:**

Affirmed.

**Final Justification:**

I maintain my original recommendation of weak accept, with moderate confidence.

The paper is well written and technically solid. The authors addressed my concerns in the rebuttal, and I consider them resolved.

Overall, I support acceptance.

**Key Questions For Authors:**

- Can you elaborate on how your work compares to another method that uses the EDL framework, namely "TrustSleepNet"?
- Line 383, column 2: "transition epochs contain features of several sleep stages". Can you please elaborate further on why transitions spark more conflict?
- Figure 2: there seems to be a relatively consistent degree of conflict around 0.4 across all epochs, why is that?
- Figure 2: around epoch 875, all views seem to be agreeing, yet, there's a short spike on the conflict degree in fig.(c) around this epoch. Can you elaborate?

**Limitations:**

yes

**Strengths And Weaknesses:**

Strengths:
- Application of evidence deep learning for multi-learning appears to be largely unexplored, and the work seems to bring novelty in that regard.
- Logical and smooth progression of the topic throughout the text.
- The experiments contain are complete and comprehensive, particularly the single individual case study is very insightful.
- Source code is available and well-documented for replication.

Weaknesses:
- A lot of details are introduced right away, and some terms are left without proper definition or introduction, such as "alignment between views", "aligned evidence extraction", "opinion versus evidence", "consistent opinions", and "hybrid category structure".
- Under the methodology section, at times it is not very clear which part is the contribution of the paper, e.g. conflict metric and the choices made for the loss function.
- The performance improvement of the method based on the metrics report appears to be marginal, but also, since many of the benchmark methods used also perform relatively highly, i.e. close to 90%, it could be that these metrics are saturated which could either mean other complementary metrics might need to be taken into account or it would be informative to look into other datasets as well.
- Clarity and readability in certain sections.

Suggestions for improvement:
- In certain sections, such as the abstract and introduction, clarity and readability of the text can be improved by providing fewer details and focusing more on the big picture.
- Some of these terms, as pointed out under weaknesses, are perhaps assumed to be known to the reader. It would be great to elaborate rather than assume. For example:
  - line 158, column 1: "consistent opinion" not properly defined earlier.
- Figure 1: caption of "evidence learning" under "Evidential DNN" is slightly misleading, since it appears that at stage f_v(.) networks only act as evidence extractors, much like feature extraction denoted under Fig. 5, but the evidence learning is the act of the whole pipeline containing evidence extraction, evidence mapping, and opinion aggregation. Please elaborate if there has been any misunderstanding regarding this.
- The theory provided mostly supports the design rationale of the method, and despite the claim of the paper, it does not validate the efficacy of the method. This matter can be slightly rephrased in the introduction and conclusion to better reflect the content.
- Significance of the results: improvements appear not large, it would be beneficial to discuss the implications of 1-5% improvement, as a general audience not so much familiar with this domain, it would be hard to judge what the ramifications of such differences are, e.g. for decision-making.

Some grammatical or typos that need attention:
- line 239, column 1: "C=0" -> "C=1"

---

> ### Author Rebuttal · Authors · 2026-03-28
>
> We sincerely appreciate your valuable and constructive comments. We have carefully addressed each comment, and our detailed responses are as follows.
>
> ---
> **Response to Weaknesses & Suggestions**
>
> We feel sorry for the weaknesses of our work and accordingly improve the manuscript based on the comment.
>
> - **Clinical significance:** We agree that the accuracy improvement of our proposed ConfSleepNet is approximately 1%–2%. Even so, such an improvement can lead to dozens of correctly classified sleep epochs over a whole night (6 to 9 hours) EEG-based sleep staging, which is highly beneficial for doctors in diagnosing various sleep disorders.
>
> - **Fig. 1 caption clarification:** We apologize for this mistake. In Fig. 1,we clarify that each evidential DNN includes feature extraction, stage-transition modeling and evidence extraction, representing the complete pipeline from input to evidence output. We employ the term “evidence learning” to denote this entire process and will provide a clearer explanation in the revised manuscript.
>
> ---
>
> **A1:** Indeed, TrustSleepNet [1] is an early work applying EDL to sleep staging. It differs from our method in two key aspects:
>
> - **Fusion strategy:** TrustSleepNet uses a single granularity for all modalities, whereas we design two granularity levels tailored to the characteristics and complementarity of EEG and EOG signals.
> - **Conflict modeling:** TrustSleepNet employs Dempster-Shafer fusion without explicit conflict modeling. In contrast, our proposed conflict-aware aggregation method explicitly accounts for conflicts among multi-view opinions. The superiority of our fusion method is demonstrated both theoretically (in _Propositions_) and experimentally.
>
> We also evaluate our proposed conflict-aware aggregation method (CMAM) on other types of classification tasks. Some results are below (full results at: https://anonymous.4open.science/r/Rebuttal-ugUx-C23C):
>
> | Dataset | EDL[2]| RCML[3]| TMCEK[4]| CMAM (Ours) |
> |-|-|-|-|-|
> | HandWritten | 97.00±0.16 | 98.70±0.19 | 97.75±0.42 | 98.45±0.58 |
> | CUB | 89.51±0.24 | 93.28±2.75 | 90.50±2.51 | 95.00±2.32 |
> | PIE | 87.99±0.56 | 93.89±2.46 | 95.15±2.81 | 95.74±1.80 |
>
> It can be seen that CMAM outperforms the existing EDL methods.
>
> ---
>
> **A2:** Higher conflict during transition arises from: (1) In terms of signal characteristics, adjacent sleep epochs during transitions contain mixed physiological features, and the differential sensitivity of EEG/EOG to these features leads to inconsistent predictions across views. (2) The model inherently has uncertainty in modeling sleep stage boundaries, and different views may capture distinct boundary information. (3) Through in-depth discussions with clinical experts, we found that inter-expert variability is most pronounced when annotating transition, reflecting the inherent difficulty of sleep stage classification, also noted in prior work[5].
>
> ---
>
> **A3:** The average conflict ~0.4 in Fig. 2(c) is an empirical observation from one MASS-SS3 subject. This value reflects the discrepancy between the belief distributions of EEG/EOG for this subject, indicating a moderate level of consistency between the two modalities. It is important to note that this baseline is not a theoretical constant ( different datasets, subjects, or modality combinations may yield different baseline conflict levels ). We will add this clarification in the revised manuscript to avoid any potential misinterpretation.
>
> ---
>
> **A4:** Around epoch 875, the predictions from different views in Fig. 2(a) and (b) appear consistent, yet the conflict degree in Fig. 2(c) exhibits a short spike. While seemingly contradictory, this phenomenon can be explained by the definition of the conflict metric.
>
> The conflict degree measures the similarity between two belief distributions rather than merely comparing their final predicted classes. Therefore, even when two opinions predict the same class, a high conflict degree can arise if their belief mass allocations differ substantially.
>
> We take $f_1$  and $f_4$  on the same instance as an example:
>
> | View | Wake | Others |
> |-|-|-|
> | $f_1$ | 0.65 | 0.30 |
> | $f_4$  | 0.50 | 0.40 |
>
> Although both views support the _W_ stage, the difference in belief distributions leads to a _C = 0.57_ computed by Eq. (6). Thus, the spike reflects the inherent asynchrony between modalities rather than inconsistency in predicted classes. We will add this explanation to the revised manuscript.
>
> ---
>
> **References**
>
> [1] Huang G et al., TrustSleepNet: A trustable deep multimodal network for sleep stage classification. BHI, 2022.
>
> [2] Sensoy M et al., Evidential deep learning to quantify classification uncertainty. NeurIPS, 2018.
>
> [3] Xu C et al.,  Reliable conflictive multi-view learning. AAAI, 2024.
>
> [4] Liang X et al., Trusted multi-view classification with expert knowledge constraints. ICML, 2025.
>
> [5] Chen X et al., ASTGSleep: Attention based spatial-temporal graph network for sleep staging. TIM, 2025.

---

> > ### Author Rebuttal · Reviewer_ugUx · 2026-04-03
> >
> > Thank you for addressing my questions; they have been fully resolved.

---

> > > ### Author Response · Authors · 2026-04-04
> > >
> > > We sincerely appreciate your thoughtful and constructive feedback. We are glad that our rebuttal addresses your key concerns, and we appreciate your positive assessment of the paper. We will revise the final version based on your suggestions.

---

### Official Review · Reviewer_MwNK · 2026-03-12

**Soundness:** 3
**Presentation:** 3
**Significance:** 2
**Originality:** 2
**Overall Recommendation:** 4
**Confidence:** 3

**Summary:**

In this paper, the authors propose ConfSleepNet, a conflict-aware evidential framework for reliable sleep stage classification using multi-modal data. The method introduces a hybrid category structure for adaptive evidence extraction and a novel conflict-aware opinion aggregation strategy that explicitly accounts for inter-view conflicts and uncertainty. Theoretical analysis and extensive experiments on multiple public datasets demonstrate that the proposed approach achieves superior performance compared to state-of-the-art methods.

**Compliance With Llm Reviewing Policy:**

Affirmed.

**Ethical Review Concerns:**

I think the author has solved my problem.  I believe the method proposed by the author has a very suitable theoretical style for ICML, and the experiments are also fully and properly conducted.

**Final Justification:**

I think the author has solved my problem.  I believe the method proposed by the author has a very suitable theoretical style for ICML, and the experiments are also fully and properly conducted.

**Key Questions For Authors:**

See pros and cons

**Limitations:**

yes

**Strengths And Weaknesses:**

Pros: 1 Theoretical analysis supports the effectiveness of the multi-opinion aggregation method.
2 The paper is well organized, well laid out, and free of technical errors.
3 The proposed ConfSleepNet effectively addresses inter-view conflicts by incorporating a conflict-aware opinion aggregation strategy, improving the robustness and reliability of sleep stage classification.
Cons: 1. Without time complexity analysis, the model may not be suitable for large-scale datasets.
2. The results of the two newer comparison algorithms lack the “F1-score for each class”.
3. Fewer algorithms are available for comparison; additional datasets or comparison algorithms may be added as appropriate.
4. Different comparison methods are used for different datasets.
5. The number of pieces of evidence used should be explained. Why were four selected?

---

> ### Author Rebuttal · Authors · 2026-03-27
>
> We sincerely appreciate your constructive suggestions, which are highly valuable for enhancing the quality of our manuscript. We have thoroughly considered all the points raised and provided our point-by-point responses below.
>
> ---
> **A1:** Thanks for your valuable suggestion. Based on this comment, we conducted a systematic complexity analysis of ConfSleepNet and compared it with multiple representative baselines.
>
> |Method|Acc (%)|Params (M)|FLOPs (G)|Training Time (s)|
> |-|-|-|-|-|
> |DeepSleepNet |77.8|23.0|17.9|259.2|
> |Dai et al. [1]|85.0|27.2|19.4|90.0|
> |FlexibleSleepNet [2]|86.8|23.3|/|82.8|
> |Phan et al. [3]|81.8|3.7|/|80.4|
> |ConfSleepNet|87.2|29.0|2.4|78.9|
>
> The results show that ConfSleepNet achieves the best Acc, maintaining comparable parameters while achieving even faster inference speed. Further, we also evaluate ConfSleepNet on varying-scale datasets, and the training time scales linearly with dataset size (Results available at: https://anonymous.4open.science/r/R1-2F4D).
>
> ---
> **A2-A4:** We sincerely appreciate your comments regarding the completeness of and fairness of experiments. Accordingly, we have conducted supplementary experiments and made modifications as follows.
>
> - **More comparison methods are added.** Based on the comment, we added several recent methods for comparison, including SleePyCo (ESWA 2024)[4], FlexibleSleepNet (JBHI 2025)[2], and HMDT-Net (IEEE TETCI 2026)[5]. Results on the two most popular sleep staging datasets are as follows, and full results please see https://anonymous.4open.science/r/R1-2F4D.
>
> **SleepEDF-20**
>
> |Method|Acc|MF1|W|N1|N2|N3|REM|
> |-|-|-|-|-|-|-|-|
> |SleePyCo|86.3|80.6|89.1|50.3|88.3|87.0|88.5|
> |FlexibleSleepNet|86.8|81.4|91.8|53.2|89.8|85.0|87.2|
> |HMDT-Net|86.4|80.5|89.1|48.5|89.2|87.3|88.4|
> |ConfSleepNet|87.2|81.8|91.9|49.5|88.2|90.4|89.1|
>
> **SleepEDF-78**
>
> |Method|Acc|MF1|W|N1|N2|N3|REM|
> |-|-|-|-|-|-|-|-|
> |SleePyCo|84.6|78.7|92.4|50.4|86.0|80.5|84.2|
> |FlexibleSleepNet|84.6|78.1|92.1|48.2|85.7|80.7|83.9|
> |HMDT-Net|84.5|77.9|92.5|50.3|84.4|81.2|80.9|
> |ConfSleepNet|85.3|78.8|93.1|45.8|86.1|82.8|86.0|
>
> It can be seen that ConfSleepNet outperforms the existing methods across all the datasets.
>
> - **Per-class F1 scores.** We feel sorry for the missing per-class F1 scores for some baselines. Following your crucial suggestion, we have supplemented all per-class F1 scores for both the original and newly added baselines.
>
> - **Test on additional datasets.** To address the issue of insufficient experimental datasets, we further evaluated the performance of ConfSleepNet on a large-scale sleep dataset, SHHS (329 subjects, 324,854 epochs)[6]. Some results are given below (full results at: https://anonymous.4open.science/r/R1-2F4D).
>
> |Method|Acc|MF1|W|N1|N2|N3|REM|
> |-|-|-|-|-|-|-|-|
> |SleePyCo|87.9|80.7|92.6|49.2|88.5|84.5|88.6|
> |FlexibleSleepNet|87.6|79.6|92.3|40.0|88.8|87.0|89.7|
> |TMCEK|84.3|78.0|90.5|43.2|87.1|85.1|83.9|
> |ConfSleepNet|88.2|81.3|93.4|44.2|90.0|89.3|89.7|
>
> The results show that ConfSleepNet also achieves the best performance on the SHHS datasets. In addition, the evaluation on this large-scale dataset confirms that ConfSleepNet is capable of operating effectively on large-scale data.
>
> ---
> **A5:** As discussed in Appendix A.1, EEG and EOG signals show complementary characteristics during sleep staging according to physiological prior knowledge. Based on this, we propose a novel design principle (Principle 1) for ConfSleepNet. Based on this principle, the four evidential DNN modules are designed as follows:
>
> - **Hybrid granularity structure.** Based on the physiological prior, we leverage EEG for fine-grained sleep stage classification (DNN $f_1$), and EOG that involves distinctive REM features for coarse-grained sleep stage classification (DNN $f_4$).
>
> - **Feature-level fusion.** Since evidence-level fusion operates at the decision level, it cannot extract interactive features of EEG and EOG at the feature level. Therefore, we further combine EEG and EOG at the feature level to capture effective features, and leverage two DNNs ($f_2$ and $f_3$) to perform fine- and coarse-grained sleep staging, respectively.
>
> ---
> The above supplementary experiments and modifications will be fully incorporated into the revision. Thank you once again for your time and expertise.
>
> ---
> **References**
>
> [1] Dai Y et al., MultichannelSleepNet: A transformer-based model for automatic sleep stage classification with PSG. IEEE JBHI, 2023.
>
> [2] Ren Z et al., FlexibleSleepNet: A model for automatic sleep stage classification based on multi-channel polysomnography. IEEE JBHI, 2025.
>
> [3] Phan et al., SleepTransformer. IEEE TBME, 2022.
>
> [4] Lee S et al., SleePyCo: automatic sleep scoring with feature pyramid and contrastive learning. ESWA, 2024.
>
> [5] Wang K et al., Heterogeneous modality dynamic trustworthy fusion network for cross-subject sleep stage classification. TETCI, 2026.
>
> [6] Quan S et al., The Sleep Heart Health Study: design, rationale, and methods. Sleep, 1997.

---

> > ### Author Rebuttal · Reviewer_MwNK · 2026-04-04
> >
> > I have no other questions. I will adjust my score appropriately based on the opinions of the other reviewers.

---

> > > ### Author Response · Authors · 2026-04-05
> > >
> > > Thank you for confirming that you have no further questions.
> > >
> > > We have carefully addressed all reviewers' concerns through additional experiments and analyses, and we are grateful to have received positive feedback.
> > >
> > > Thank you for your time and constructive suggestions, which have been invaluable to our work.

---

### Decision · Program_Chairs · 2026-04-30

**Decision:**

Accept (regular)

**Comment:**

In this paper, the authors propose ConfSleepNet, a conflict-aware evidential framework for reliable sleep stage classification using multi-modal data. The method introduces a hybrid category structure for adaptive evidence extraction and a novel conflict-aware opinion aggregation strategy that explicitly accounts for inter-view conflicts and uncertainty. The paper is in a good quality, and all of the issues are well responsed. I would recommend the authors carefully revise the paper in the final version.